# xLSTM-Mixer: Multivariate Time Series Forecasting by Mixing via Scalar Memories

**Maurice Kraus**[1,*]     **Felix Divo**[1,*]     **Devendra Singh Dhami**[2]     **Kristian Kersting**[1,3,4,5]

[1]AI & ML Group, TU Darmstadt     [2]TU Eindhoven     [3]Hessian Center for AI (hessian.AI)
[4]German Research Center for AI (DFKI)     [5]Centre for Cognitive Science, TU Darmstadt
{maurice.kraus,felix.divo,kersting}@cs.tu-darmstadt.de     d.s.dhami@tue.nl

## Abstract

Time series data is prevalent across numerous fields, necessitating the development of robust and accurate forecasting models. Capturing patterns both within and between temporal and multivariate components is crucial for reliable predictions. We introduce xLSTM-Mixer, a model designed to effectively integrate temporal sequences, joint time-variate information, and multiple perspectives for robust forecasting. Our approach begins with a linear forecast shared across variates, which is then refined by xLSTM blocks. They serve as key elements for modeling the complex dynamics of challenging time series data. xLSTM-Mixer ultimately reconciles two distinct views to produce the final forecast. Our extensive evaluations demonstrate its superior long-term forecasting performance compared to recent state-of-the-art methods while requiring very little memory. A thorough model analysis provides further insights into its key components and confirms its robustness and effectiveness. This work contributes to the resurgence of recurrent models in forecasting by combining them, for the first time, with mixing architectures.

## 1 Introduction

Time series are an essential data modality ubiquitous in many critical fields of application, such as medicine [Hosseini et al., 2021], manufacturing [Essien and Giannetti, 2020], logistics [Seyedan and Mafakheri, 2020], traffic management [Lippi et al., 2013], finance [Lin et al., 2012, Divo et al., 2025], and weather modeling [Lam et al., 2023]. While significant progress in time series forecasting has been made over the decades, the field is still far from being solved. Further increasing the forecast quality obtained from machine learning models promises a manifold of improvements, such as more accurate medical treatments, increased efficiency in manufacturing and transportation, and higher crop yields.

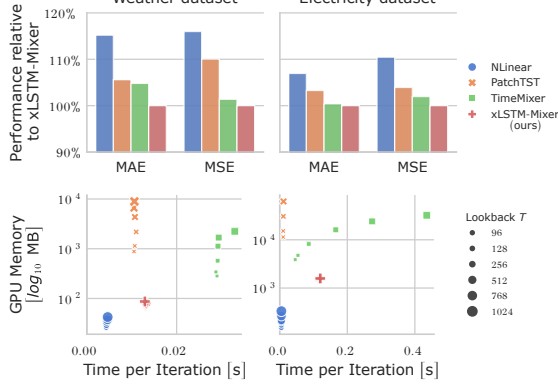

Figure 1: **xLSTM-Mixer provides excellent forecasts with a very low memory footprint while being sufficiently fast.** Details are found in Sec. 4.4.

---

[*]Authors contributed equally.

39th Conference on Neural Information Processing Systems (NeurIPS 2025).

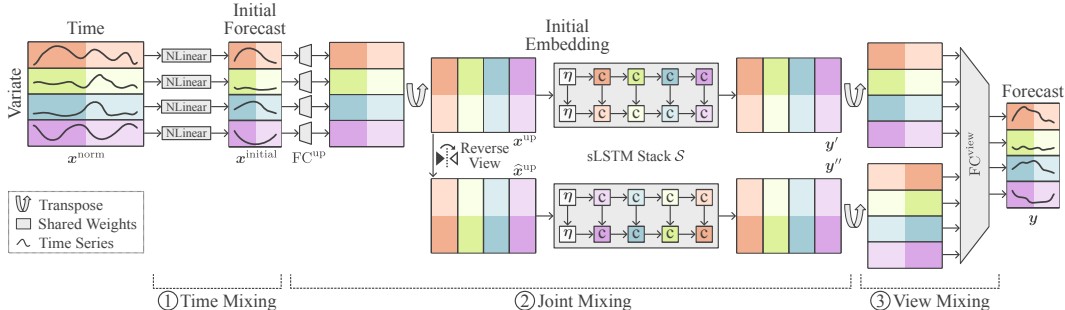

Figure 2: **The xLSTM-Mixer architecture consists of three stages:** (1) An initial NLinear forecast assuming channel independence and performing *time mixing*; (2) subsequent *joint mixing*, which mixes variate and time information through crucial applications of sLSTM blocks; and (3) *view mixing*, where the two latent forecast views are reconciled into a coherent final forecast.

Historically, recurrent neural networks (RNNs) and their powerful successors were natural choices for deep learning-based time series forecasting [Hochreiter and Schmidhuber, 1997, Cho et al., 2014]. Today, large Transformers [Vaswani et al., 2017] are applied extensively to time series tasks, including forecasting. Many improvements to the vanilla architecture have since been proposed, including patching [Nie et al., 2023], decompositions [Zeng et al., 2023], and tokenization inversions [Liu et al., 2023]. Newer approaches include pretrained models, which, however, usually cannot capture relationships between multiple variables [Ansari et al., 2024]. Yet, some fundamental limitations of Transformers are yet to be lifted. For instance, they are inefficient when applied to long sequences due to the cost of the attention mechanism being quadratic in the number of variates and time steps. As embedded devices and edge computing platforms are gaining importance, the demand for lightweight forecasting models that balance accuracy with minimal memory and computational overhead grows. Therefore, recurrent and state space models (SSMs) [Patro and Agneeswaran, 2025] are experiencing a resurgence of interest in overcoming such limitations. Specifically, Beck et al. [2024] revisited recurrent models by borrowing insights gained from Transformers in many domains, specifically natural language processing. They propose Extended Long Short-Term Memory (xLSTM) models as alternatives to current sequence models.

We propose xLSTM-Mixer[1], a new state-of-the-art method for time series forecasting using recurrent deep learning methods, which strikes a balance by providing strong forecasting accuracy while remaining highly efficient, as Fig. 1 shows. Architecturally, we combine the highly expressive xLSTM architecture with carefully crafted time, variate, and multi-view mixing. These operations regularize the training and limit the number of model parameters by weight-sharing, effectively improving the learning of features necessary for accurate forecasting. xLSTM-Mixer initially computes a channel-independent linear forecast shared over the variates. It is then up-projected to a higher hidden dimension and subsequently refined by an xLSTM stack. It performs multi-view forecasting by producing a forecast from the original and reversed up-projected embedding. The powerful xLSTM cells thereby jointly mix time and variate information to capture complex patterns from the data. Both forecasts are eventually reconciled by a learned linear projection into the final prediction, called view mixing. An overview of our method is shown in Fig. 2.

Overall, we make the following contributions:

  (i) We investigate time and variate mixing in the context of recurrent models and propose a joint multistage approach that is highly effective for multivariate time series forecasting. We argue that marching over the variates instead of the temporal axis yields better results if suitably combined with temporal mixing.

 (ii) We propose xLSTM-Mixer, a state-of-the-art method for time series forecasting, for the first time combining recurrent deep learning with a mixing architecture.

(iii) We extensively compare xLSTM-Mixer with existing methods for multivariate long-term time series forecasting and perform in-depth model analyses. The experiments demonstrate that xLSTM-Mixer consistently excels in a wide range of benchmarks.

---

[1] Code available at `https://github.com/mauricekraus/xlstm-mixer`

**Outline.** In the upcoming Sec. 2, we introduce preliminaries to allow us to motivate and explain xLSTM-Mixer in Sec. 3. We then comprehensively evaluate its effectiveness and inner workings in Sec. 4. We finally review related work in Sec. 5 and close with a conclusion and outlook in Sec. 6.

## 2 Background

After introducing the notation used throughout this work, we review xLSTM blocks and discuss whether leveraging channel mixing or their independence is beneficial in time series models.

**Notation.** In multivariate time series forecasting, the model is presented with a time series $\boldsymbol{X} = (\boldsymbol{x}_1, \ldots, \boldsymbol{x}_T) \in \mathbb{R}^{V \times T}$ consisting of $T$ time steps with $V$ variates each. Given this context, the forecaster shall predict the future values $\boldsymbol{Y} = (\boldsymbol{x}_{T+1}, \ldots, \boldsymbol{x}_{T+H}) \in \mathbb{R}^{V \times H}$ up to a horizon $H$. A variate (sometimes called a channel) can be any scalar measurement, such as the occupancy of a road or the temperature in a power plant. The measurements are assumed to be carried out jointly, such that the $T + H$ time steps reflect a regularly sampled signal. A time series dataset consists of $N$ such pairs $\left\{ \left( \boldsymbol{X}^{(i)}, \boldsymbol{Y}^{(i)} \right) \right\}_{i \in \{1, \ldots, N\}}$ divided into train, validation, and test portions.

### 2.1 Extended Long Short-Term Memory (xLSTM)

Beck et al. [2024] propose xLSTM consisting of two building blocks, namely the sLSTM and mLSTM modules. To harness the full expressivity of xLSTMs within each step and across the computation sequence, we employ a stack of sLSTM blocks without any mLSTM blocks. The latter are less suited for joint mixing due to their independent treatment of the sequence elements, making it impossible to learn any relationships between them directly. See App. B for a deeper discussion. We will continue by recalling how sLSTM cells function. The standard LSTM architecture of Hochreiter and Schmidhuber [1997] involves updating the cell state $\boldsymbol{c}_t$ through a combination of input, forget, and output gates regulating the flow of information across tokens. sLSTM blocks enhance this by incorporating exponential gating and memory mixing [Greff et al., 2017] to handle complex temporal and cross-variate dependencies better. The sLSTM updates the cell $\boldsymbol{c}_t$ and hidden state $\boldsymbol{h}_t$ as follows:

$$\boldsymbol{c}_t = \boldsymbol{f}_t \odot \boldsymbol{c}_{t-1} + \boldsymbol{i}_t \odot \boldsymbol{z}_t \qquad\qquad\qquad \text{cell state} \quad (1)$$

$$\boldsymbol{n}_t = \boldsymbol{f}_t \cdot \boldsymbol{n}_{t-1} + \boldsymbol{i}_t \qquad\qquad\qquad \text{normalizer state} \quad (2)$$

$$\boldsymbol{h}_t = \boldsymbol{o}_t \odot \boldsymbol{c}_t \odot \boldsymbol{n}_t^{-1} \qquad\qquad\qquad \text{hidden state} \quad (3)$$

$$\boldsymbol{z}_t = \tanh\left(\boldsymbol{W}_z \boldsymbol{x}_t + \boldsymbol{R}_z \boldsymbol{h}_{t-1} + \boldsymbol{b}_z\right) \qquad\qquad\qquad \text{cell input} \quad (4)$$

$$\boldsymbol{i}_t = \exp\left(\tilde{\boldsymbol{i}}_t - \boldsymbol{m}_t\right) \qquad \tilde{\boldsymbol{i}}_t = \boldsymbol{W}_i \boldsymbol{x}_t + \boldsymbol{R}_i \boldsymbol{h}_{t-1} + \boldsymbol{b}_i \qquad \text{input gate} \quad (5)$$

$$\boldsymbol{f}_t = \exp\left(\tilde{\boldsymbol{f}}_t + \boldsymbol{m}_{t-1} - \boldsymbol{m}_t\right) \qquad \tilde{\boldsymbol{f}}_t = \boldsymbol{W}_f \boldsymbol{x}_t + \boldsymbol{R}_f \boldsymbol{h}_{t-1} + \boldsymbol{b}_f \qquad \text{forget gate} \quad (6)$$

$$\boldsymbol{o}_t = \sigma\left(\boldsymbol{W}_o \boldsymbol{x}_t + \boldsymbol{R}_o \boldsymbol{h}_{t-1} + \boldsymbol{b}_o\right) \qquad\qquad\qquad \text{output gate} \quad (7)$$

$$\boldsymbol{m}_t = \max\left(\tilde{\boldsymbol{f}}_t + \boldsymbol{m}_{t-1}, \tilde{\boldsymbol{i}}_t\right) \qquad\qquad\qquad \text{stabilizer state} \quad (8)$$

In this setup, the matrices $\boldsymbol{W}_z, \boldsymbol{W}_i, \boldsymbol{W}_f$, and $\boldsymbol{W}_o$ are input weights mapping the input token $\boldsymbol{x}_t$ to the cell input $\boldsymbol{z}_t$, input gate, forget gate, and output gate, respectively. The states $\boldsymbol{n}_t$ and $\boldsymbol{m}_t$ serve as necessary normalization and training stabilization, respectively. As Beck et al. have shown, it is beneficial to restrict the memory mixing performed by the recurrent weight matrices $\boldsymbol{R}_z, \boldsymbol{R}_i, \boldsymbol{R}_f$, and $\boldsymbol{R}_o$ to individual *heads*, inspired by the multi-head setup of Transformers [Zeng et al., 2023], yet more restricted and efficient. In particular, each token gets broken up into groups of features, where the input weights $\boldsymbol{W}_{z,i,f,o}$ act across all of them, but the recurrence matrices $\boldsymbol{R}_{z,i,f,o}$ are implemented as block-diagonal. This permits specialization of the individual heads to patterns specific to the respective section of the tokens and empirically does not sacrifice expressivity.

### 2.2 Channel Independence and Mixing for Time Series

Multiple works have investigated whether it is beneficial to learn representations of the time and variate dimensions jointly or separately. Intuitively, because joint mixing is strictly more expressive, one might think it should always be preferred. And, indeed, it is used in many works including Temporal Convolutional Networks [Lea et al., 2016], N-BEATS [Oreshkin et al., 2019], N-HiTS [Challu et al., 2023], and many Transformers [Lim et al., 2021, Wu et al., 2021, Zhou et al., 2022]. However,

treating slices of the input data independently assumes an invariance to temporal or variate positions and serves as a strong regularization against overfitting, reminiscent of kernels in CNNs. Prominent models implementing some aspects of channel independence in multivariate time series forecasting are PatchTST [Nie et al., 2023] and iTransformer [Liu et al., 2023]. TiDE [Das et al., 2023], on the other hand, contains a time-step shared feature projection and temporal decoder but treats variates jointly. As Tolstikhin et al. [2021] have shown with MLP-Mixer, interleaving mixing of all channels per token and all tokens per channel does not empirically sacrifice any expressivity and instead improves performance and efficiency. This idea has since been applied to time series too, namely in architectures such as TimeMixer(++) [Wang et al., 2024a, 2025a], TSMixer [Chen et al., 2023c], WPMixer [Murad et al., 2025], and PMformer [Lee et al., 2024], and is, therefore, one key component of our method xLSTM-Mixer.

## 3    xLSTM-Mixer

We now explain xLSTM-Mixer shown in Fig. 2 in more detail. It carefully integrates several key components: an initial linear forecast with time mixing, joint mixing using powerful sLSTM blocks, and an eventual combination of two views by a final fully connected layer. The transposing steps between the components enable capturing complex temporal and intra-variate patterns while facilitating easy trainability and limiting parameter counts. The sLSTM blocks, in particular, can learn intricate non-linear relationships hidden within the data along both the time and variate dimensions. The architecture is furthermore equipped with normalization layers and skip connections to improve training stability and overall effectiveness.

### 3.1    Normalization and Initial Linear Forecast

Normalization has become an essential ingredient of modern deep learning architectures [Huang et al., 2023]. For time series in particular, reversible instance norm (RevIN) [Kim et al., 2022] is a general recipe for improving forecasting performance, where each time series instance is normalized by its mean and variance and furthermore scaled and offset by a learnable scale $\boldsymbol{\gamma}$ and offset $\boldsymbol{\beta}$:

$$\boldsymbol{x}_t^{\text{norm}} = \text{RevIN}(\boldsymbol{x}_t) = \boldsymbol{\gamma} \odot \left( \frac{\boldsymbol{x}_t - \mathbb{E}\left[\boldsymbol{x}\right]}{\sqrt{\text{Var}\left[\boldsymbol{x}\right] + \boldsymbol{\epsilon}}} \right) + \boldsymbol{\beta}.$$

We apply it as part of xLSTM-Mixer, and at the end of the entire pipeline, we invert the RevIN operation to obtain the final prediction. In the case of xLSTM-Mixer, the typical skip connections found in mixer acrchitectures [Tolstikhin et al., 2021, Chen et al., 2023c] are taken up by RevIN, the normalization in the NLinear forecast [Zeng et al., 2023], and the integral skip connections within each sLSTM block.

It has been shown previously that simple linear models equipped with appropriate normalization schemes are, already by themselves, decent long-term forecasters [Zeng et al., 2023, Li et al., 2023]. Our observations confirm this finding. Therefore, we first process each variate separately by an NLinear model by computing:

$$\boldsymbol{x}^{\text{initial}} = \text{NLinear}(\boldsymbol{x}^{\text{norm}}) = \text{FC}\left(\boldsymbol{x}_{1:T}^{\text{norm}} - x_T^{\text{norm}}\right) + x_T^{\text{norm}},$$

where $\text{FC}(\cdot) : \mathbb{R}^T \to \mathbb{R}^H$ denotes a fully-connected linear layer with bias term. Sharing this model across variates limits parameter counts, and the weight-tying serves as a useful regularization. The quality of this initial forecast will be investigated in Sec. 4.1 and 4.5.

### 3.2    sLSTM Refinement

While the NLinear forecast $\boldsymbol{x}^{\text{initial}} \in \mathbb{R}^{V \times H}$ captures the basic patterns between the historic and future time steps, its quality alone is insufficient for today's challenging datasets. We, therefore, refine it using powerful sLSTM blocks. As a first step, it is crucial to increase the embedding dimension of the data to provide sufficient latent dimensions $D$ for the sLSTM cells as $\boldsymbol{x}^{\text{up}} = \text{FC}^{\text{up}}\left(\boldsymbol{x}^{\text{initial}}\right)$. This prior up-projection is similar to what is commonly performed in SSMs [Beck et al., 2024]. We weight-share $\text{FC}^{\text{up}} : \mathbb{R}^H \to \mathbb{R}^D$ across variates to perform time-mixing similar to the initial forecast. Note that this step does not maintain the temporal ordering within the embedding token dimensions, as was the case up until this step, and instead embeds it into a higher latent dimension.

The stack of $M$ sLSTM blocks $\mathcal{S}(\cdot)$ transforms $\boldsymbol{x}^{\text{up}} \in \mathbb{R}^{V \times D}$ as defined in Eq. 1 to 8. The recurrent model strides over the data in variate order, i.e., where each token represents all time steps from a single variate as in the work of Liu et al. [2023]. The sLSTM blocks learn intricate non-linear relationships hidden within the data along both the time and variate dimensions. The mixing of the hidden state is still limited to blocks of consecutive dimensions, aiding efficient learning and inference while allowing for effective cross-variate interaction during the recurrent processing. Striding over variates has the benefit of linear time scaling in the number of variates at a constant number of parameters. It, however, comes at the cost of possibly fixing a suboptimal order of variates. While this is empirically not a significant limitation (see Sec. 4.4), we leave investigations into how to find a suitable ordering for future work. In addition to a large embedding dim, we observed a high number of heads being crucial for effective forecasting.

The sLSTM cells' first hidden state $\boldsymbol{h}_0$ must be initialized before each sequence of tokens can be processed. Extending the initial description of these blocks, we propose learning a single initial embedding token $\boldsymbol{\eta} \in \mathbb{R}^D$ that gets prepended to each encoded time series $\boldsymbol{x}^{\text{up}}$. These initial embeddings draw from recent advances in Large Language Models, where learnable "soft prompt" tokens are used to condition models and improve their ability to generate coherent outputs [Lester et al., 2021, Li and Liang, 2021, Chen et al., 2023a,b]. Recent research has extended the application of soft prompts to LLM-based time series forecasting [Cao et al., 2023, Sun et al., 2024, Jin et al., 2024], emphasizing their adaptability and effectiveness in improving model performance across modalities. These tokens enable greater flexibility by conditioning its initial memory representation to specific dataset characteristics for dynamically interacting with the time and variate data. Soft prompts can be readily optimized through back-propagation with very little overhead.

### 3.3 Multi-View Mixing

To further regularize the training of the sLSTM as with the linear projections, we compute forecasts from the original embedding $\boldsymbol{x}^{\text{up}}$ as well as the reversed embedding $\widehat{\boldsymbol{x}}^{\text{up}}$, where the order of the latent dimensions including the representation of $\boldsymbol{\eta}$ is inverted. Learning forecasts $\boldsymbol{y}', \boldsymbol{y}'' \in \mathbb{R}^{V \times D}$ for both views while sharing weights helps learn better representations. Such multi-task learning settings are known to benefit training [Zhang and Yang, 2022]. It can also be viewed as ensembling over different variate orderings with weight sharing. The final forecast is obtained by a linear projection $\text{FC}^{\text{view}} : \mathbb{R}^D \times \mathbb{R}^D \to \mathbb{R}^H$ of the two forecasts, again per-variate. Specifically, we compute $\boldsymbol{y}^{\text{norm}} = \text{FC}^{\text{view}}(\boldsymbol{y}', \boldsymbol{y}'')$, where $\boldsymbol{y}' = \mathcal{S}(\boldsymbol{x}^{\text{up}})$ and $\boldsymbol{y}'' = \mathcal{S}(\widehat{\boldsymbol{x}}^{\text{up}})$. The final forecast is obtained after de-normalizing the reconciled forecasts as $\boldsymbol{y} = \text{RevIN}^{-1}(\boldsymbol{y}^{\text{norm}})$.

## 4 Experiments

We conduct a series of experiments to evaluate the forecasting capabilities of xLSTM-Mixer, aiming to provide comprehensive insights into its performance. Our primary focus is on long-term forecasting, following the works of Das et al. [2023], Chen et al. [2023c], Lin et al. [2024], and Liu et al. [2025]. Further tasks are explored in Sec. 4.3. Additionally, we perform an extensive model analysis, including visualizations of the initial embedding tokens, hyperparameter sensitivity, and performance measurement. Finally, an ablation study identifies the contributions of the individual components.

**Datasets.** We generally follow the established benchmark procedure of Wu et al. [2021] and Zhou et al. [2021] for best backward and future comparability. The datasets we thus used are provided as an overview in App. D. **Training.** We follow standard practice in the forecasting literature by evaluating long-term forecasts using mean squared error (MSE) and mean absolute error (MAE). Based on our experiments, we used MAE as the training loss function since it yielded the best results. The datasets were standardized for consistency across features. In addition, we conducted all experiments three times and reported the averaged values. Further details on hyperparameter selection, metrics, and implementation can be found in App. C. **Baseline Models.** We compare xLSTM-Mixer to the recurrent models xLSTMTime [Alharthi and Mahmood, 2024] and LSTM [Hochreiter and Schmidhuber, 1997]; mixer models TimeMixer++ [Wang et al., 2025a], TimeMixer [Wang et al., 2024a], and TSMixer [Chen et al., 2023c]; MLP-based models CycleNet/MLP [Lin et al., 2024], DLinear [Zeng et al., 2023], and TiDE [Das et al., 2023]; the SSMs S-Mamba [Wang et al., 2025b] and Chimera [Behrouz et al., 2024]; the Transformers PatchTST [Nie et al., 2023] and iTransformer [Liu et al., 2023]; the convolutional architectures ModernTCN [Donghao and Xue,

Table 1: **xLSTM-Mixer is effective in long-term forecasting.** The results are averaged from 4 different prediction lengths {96, 192, 336, 720}. A lower MSE or MAE indicates a better prediction. The **best** result for each dataset is highlighted bold red and second-best blue and underlined. Wins for each model out of all 28 settings are shown at the bottom. Full results are provided in App. E.

| Models | Recurrent | | | | | | Mixer | | | | | | MLP | | | | | | State Space | | | | Trans. | | Conv. | | Pretrained* | | | |
|---|---|---|---|---|---|---|---|---|---|---|---|---|---|---|---|---|---|---|---|---|---|---|---|---|---|---|---|---|---|---|
| | xLSTM-Mixer | | xLSTMTime 2024 | | LSTM 1997† | | TimeMix.++ 2025a | | TimeMix. 2024a | | TSMixer 2023c | | CycleNet 2024 | | DLinear 2023 | | TiDE 2023 | | S-Mamba 2025b | | Chimera 2024 | | PatchTST 2023 | | Mod.TCN 2023 | | Timer-XL 2025 | | Moirai$_{Base}$ 2024 | |
| Dataset | MSE | MAE | MSE | MAE | MSE | MAE | MSE | MAE | MSE | MAE | MSE | MAE | MSE | MAE | MSE | MAE | MSE | MAE | MSE | MAE | MSE | MAE | MSE | MAE | MSE | MAE | MSE | MAE | MSE | MAE |
| Weather | **0.219** | **0.250** | 0.222 | 0.255 | 0.444 | 0.454 | 0.226 | 0.262 | 0.222 | 0.262 | 0.225 | 0.264 | 0.223 | 0.264 | 0.246 | 0.300 | 0.236 | 0.282 | 0.251 | 0.276 | **0.219** | 0.258 | 0.241 | 0.264 | 0.224 | 0.264 | 0.256 | 0.294 | 0.287 | 0.281 |
| Electricity | **0.153** | **0.245** | 0.157 | 0.250 | 0.559 | 0.549 | 0.165 | 0.253 | 0.156 | 0.246 | 0.160 | 0.256 | 0.156 | 0.251 | 0.166 | 0.264 | 0.159 | 0.257 | 0.170 | 0.265 | 0.154 | 0.249 | 0.159 | 0.253 | 0.156 | 0.253 | 0.174 | 0.278 | 0.187 | 0.274 |
| Traffic | 0.392 | **0.253** | 0.391 | 0.261 | 1.011 | 0.541 | 0.416 | 0.264 | 0.387 | 0.262 | 0.408 | 0.284 | 0.403 | 0.282 | 0.434 | 0.295 | **0.356** | 0.261 | 0.414 | 0.276 | 0.403 | 0.286 | 0.391 | 0.264 | 0.396 | 0.270 | –‡ | – | –‡ | – |
| ETTh1 | **0.397** | 0.420 | 0.408 | 0.428 | 1.198 | 0.821 | 0.419 | 0.432 | 0.411 | 0.423 | 0.412 | 0.428 | 0.435 | 0.440 | 0.423 | 0.437 | 0.419 | 0.430 | 0.455 | 0.450 | 0.405 | 0.424 | 0.413 | 0.434 | 0.404 | 0.420 | 0.404 | **0.417** | 0.417 | 0.419 |
| ETTh2 | 0.340 | 0.382 | 0.346 | 0.386 | 3.095 | 1.352 | 0.339 | 0.380 | **0.316** | 0.384 | 0.355 | 0.401 | 0.367 | 0.405 | 0.431 | 0.447 | 0.345 | 0.394 | 0.381 | 0.405 | 0.318 | **0.375** | 0.324 | 0.381 | 0.322 | 0.379 | 0.347 | 0.388 | 0.362 | 0.382 |
| ETTm1 | **0.339** | **0.366** | 0.347 | 0.372 | 1.142 | 0.782 | 0.369 | 0.378 | 0.348 | 0.375 | 0.347 | 0.375 | 0.360 | 0.388 | 0.357 | 0.379 | 0.355 | 0.378 | 0.398 | 0.405 | 0.345 | 0.377 | 0.353 | 0.382 | 0.351 | 0.381 | 0.373 | 0.392 | 0.406 | 0.385 |
| ETTm2 | **0.248** | **0.307** | 0.254 | 0.310 | 2.395 | 1.177 | 0.269 | 0.320 | 0.256 | 0.315 | 0.267 | 0.322 | 0.263 | 0.324 | 0.267 | 0.332 | 0.249 | 0.312 | 0.288 | 0.332 | 0.250 | 0.316 | 0.256 | 0.317 | 0.253 | 0.314 | 0.273 | 0.336 | 0.311 | 0.337 |
| Wins | **11** | **16** | 0 | 2 | 0 | 0 | 0 | 2 | 2 | 2 | 0 | 0 | 1 | 0 | 0 | 0 | 5 | 1 | 0 | 0 | 8 | 3 | 1 | 0 | 2 | 3 | 0 | 1 | 0 | 0 |

* Zero-shot forecasting. † Taken from Wu et al. [2022a]. ‡ Traffic/PEMS are often used during pre-training [Liu et al., 2025]. Thus, no zero-shot results are available.

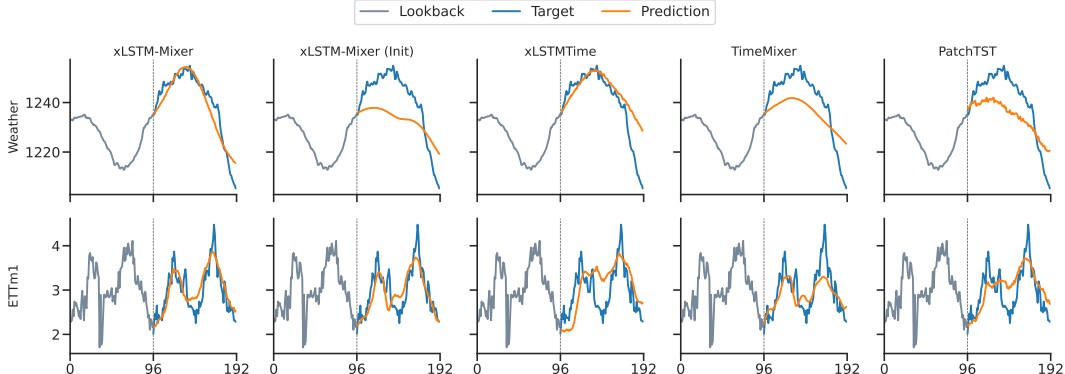

Figure 3: **xLSTM-Mixer provides convincing forecasts.** This figure shows example forecasts on the Weather and ETTm1 datasets for multiple models with lookback windows and forecasting horizons fixed at 96. The first panel illustrates the forecast from xLSTM-Mixer, while the second shows the initial forecast extracted before the up-projection step, highlighting the effectiveness of our added components. Comparisons with further baselines are provided for context.

2023] and TimesNet [Wu et al., 2022a]; and the pretrained zero-shot forecasters Timer-XL [Liu et al., 2025] and Moirai$_{Base}$ [Woo et al., 2024]. **On choosing lookback lengths $L$.** Some prior works on long-term forecasting fix the lookback window $L$ for the sake of a fair comparison. However, Abdelmalak et al. [2025] and Brigato et al. [2025] argue that fixing especially to 96, which is common in today's benchmarks, can substantially distort the comparisons. When allowed to tune each model's input length, most baselines improve and simple linear or MLP backbones close much of the gap to Transformers, whereas at fixed $L = 96$ they appear artificially weak. We agree with these findings and therefore provide results for optimal hyperparameters for all baselines and xLSTM-Mixer in our experiments. See also Fig. 4b in Sec. 4.4 for a sensitivity analysis on $L$.

## 4.1 Long-Term Time Series Forecasting

We present the performance of xLSTM-Mixer compared to prior models in Tab. 1. The full results and standard deviations are provided in App. E. As shown, xLSTM-Mixer consistently delivers highly accurate forecasts across a wide range of datasets. It achieves the best results in 11 out of 28 cases for MSE and 16 out of 28 cases for MAE, demonstrating its superior performance in long-term forecasting. xLSTM-Mixer defines a new state-of-the-art on six out of seven datasets. This shows that xLSTM-Mixer consistently delivers excellent performance in long-term forecasting. A qualitative comparison with several baselines, including the initial forecast extracted before the sLSTM refinement, is shown in Fig. 3. Here, both the lookback window and the forecasting horizon are fixed at 96.

Table 2: **xLSTM-Mixer excels on GIFT-Eval.** The table reports the top 10 across all categories.

| Model | MASE ↓ | CRPS ↓ | Rank (CRPS) ↓ |
|---|---|---|---|
| TiRex | 0.724 | 0.498 | 1 |
| **xLSTM-Mixer (ours)** | 0.780 | 0.510 | 2 |
| TEMPO_ensemble | 0.862 | 0.514 | 3 |
| Toto_Open_Base_1.0 | 0.750 | 0.517 | 4 |
| TabPFN-TS | 0.771 | 0.544 | 5 |
| YingLong_300m | 0.798 | 0.548 | 6 |
| timesfm_2_0_500m | 0.758 | 0.550 | 7 |
| YingLong_110m | 0.809 | 0.557 | 8 |
| sundial_base_128m | 0.750 | 0.559 | 9 |
| YingLong_50m | 0.822 | 0.567 | 10 |

**Significance Testing.** In addition to the common practice of evaluating by counting wins across models and datasets, we perform statistical testing following the well-known practice of Demšar [2006]. First, a Friedman test ensures the model's performances follow different distributions ($p < 10^{-10}$). We can then perform a Conover post-hoc test while adjusting for the family-wise error rate using Holm's method. We must restrict this comparison to one metric (MSE) and horizon (96 steps), to avoid the strongly correlated results for different horizons to artificially inflate significance levels. At a significance threshold of $p = 0.05$, we find that xLSTM-Mixer is statistically significantly better than all other methods, except for xLSTMTime. Yet, the difference in average rank is still impressive, with 1.5 for xLSTM-Mixer and 4.0 for xLSTMTime. Furthermore, xLSTMTime is statistically inseparable from many other models, namely TimeMixer, TSMixer, ModernTCN, Chimera, PatchTST, and TiDE.

## 4.2 Forecasting on the GIFT-Eval Benchmark

To showcase xLSTM-Mixer's versatility beyond multivariate long-term point forecasting, we evaluate it on GIFT-Eval [Aksu et al., 2024]. It is a comprehensive benchmark for general time-series forecasting spanning diverse domains, frequencies, horizons, and both univariate and multivariate regimes. The benchmark emphasizes probabilistic assessment and supports standardized (including zero-shot) evaluation protocols. Following the official setup, we report aggregated Mean Absolute Scaled Error (MASE) and Continuous Ranked Probability Score (CRPS), where lower is better. To extend xLSTM-Mixer to probabilistic forecasting capabilities, we equip it with a head predicting quantiles, yielding full predictive distributions rather than point estimates.

Results are summarized in Tab. 2 and detailed in App. F. By CRPS, xLSTM-Mixer is the top purely supervised model (i.e., without using the GIFT-Eval pretraining corpus) and ranks 2nd overall. This demonstrates that although xLSTM-Mixer is designed for multivariate forecasting, it remains highly competitive among methods across univariate and multivariate settings and for both short and long horizons in the probabilistic regime.

**Significance Testing.** As in the previous Sec. 4.1, we rigorously test for the significance of the improvement of xLSTM-Mixer over the existing methods. After the initial Firedman test on the MSE, we see that xLSTM-Mixer is significantly better than all other 38 methods, except for sundial_base_128m, TiRex, and TTM-R2-Finetuned. See Fig. 5 in App. F for a complete critical difference diagram.

## 4.3 Outlook on Classification

To illustrate performance beyond forecasting, we evaluate xLSTM-Mixer on time-series classification by using it as an embedding model. For this, we replace the final projection layer with a single classification head. xLSTM-Mixer achieves very strong performance on standard time-series classification benchmarks (cf. App. J). These findings suggest that approaches reconciling recurrent and mixing architectures, such as xLSTM-Mixer, are highly flexible yet powerful.

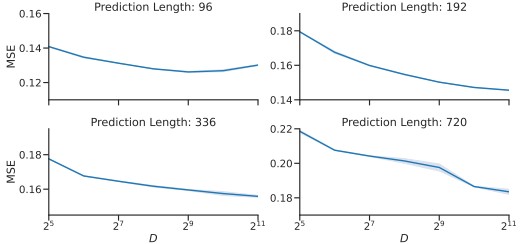 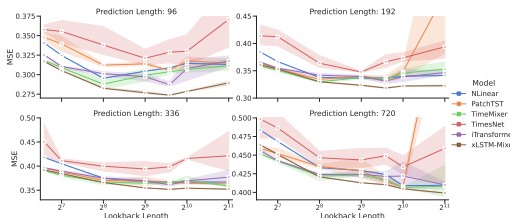

(a) **Increasing the hidden dimension $D$ becomes increasingly beneficial at longer prediction lengths $H$.** Shown is the MSE for varying $D$ on the Electricity dataset.

(b) **Increasing the lookback window $T$ increases forecasting performance,** with xLSTM-Mixer virtually always providing the best results. Shows the MSE ($\downarrow$) on the ETTm1 dataset.

Figure 4: Impact of model parameters on forecasting performance.

## 4.4 Model Analysis

**Initial Token Embeddings.** We qualitatively inspect decodings of the initial embedding tokens $\boldsymbol{\eta}$ on multiple datasets to further understand and interpret the initializations learned by xLSTM-Mixer. $\boldsymbol{\eta}$ are decoded to a forecast $\boldsymbol{y}$ by transforming them through the sLSTM stack $\mathcal{S}$ and applying multi-view mixing. The resulting output of $\mathrm{FC}^{\mathrm{view}}$ can then be interpreted as the conditioning time series used to initialize the sLSTM blocks. Fig. 6 in Appendix App. G shows the dataset-specific patterns the initial embedding tokens have learned for various horizons $H$. Increasing prediction horizons eventually reveal underlying seasonal patterns and respective dataset dynamics.

**Model Efficiency.** To assess the computational resources required for using xLSTM-Mixer, we measured the average wall-clock time and peak graphics card memory required to perform a training step. Fig. 1 shows how this changes over multiple lookback lengths $T$ and two datasets at a forecast horizon of $H = 336$. Compared to baselines, xLSTM-Mixer scales extremely favorably in $T$, only exhibiting a negligible increase in time and memory requirements. While computations take slightly longer for larger lookback sizes, the increase is much smaller than for Transformer-based models. One advantage of TimeMixer was its efficiency over Transformers, upon which xLSTM-Mixer now significantly improves by requiring one to two orders of magnitude less memory.

**Sensitivity to the xLSTM Hidden Dimension.** In Fig. 4a, we visualize the performance of xLSTM-Mixer on the Electricity dataset with increasing sLSTM embedding hidden dimension $D$ realized by $\mathrm{FC}^{\mathrm{up}}$. The results indicate that a larger $D$ enables xLSTM-Mixer to better capture the complexity of the series over extended horizons, leading to improved forecasting accuracy.

**Robustness to the Lookback Length.** Fig. 4b illustrates the performance of xLSTM-Mixer across varying lookback lengths $T$ and prediction horizons $H$. Note that we had to rerun some experiments for TimeMixer at $T = 720$ with varying seeds since many training runs diverged. We observe that xLSTM-Mixer can effectively utilize longer lookback windows than the baselines, especially when compared to transformer-based models. This advantage stems from xLSTM-Mixer's avoidance of self-attention, allowing it to handle extended lookback lengths efficiently. On short prediction lengths with $T \in \{96, 192\}$, information of more than 768 time steps in the past becomes redundant to inform the comparatively short forecast, causing models to deteriorate slightly. On longer horizons, increasingly farther lookbacks become useful for forecasting. Additionally, xLSTM-Mixer demonstrates stable and consistent performance with low variance across scales.

**xLSTM-Mixer Captures Cross-Variate Patterns.** To determine if our chosen ordering of variates is viable, we conduct additional experiments investigating the impact of different variate permutations in Tab. 12 in App. H. While we observe that the performance does depend to a certain extent on the ordering of variates, this does not pose a significant limitation in practice since the standard ordering provided with each dataset already yields strong forecasts. Moreover, our attribution analysis (cf. Fig. 7) confirms that xLSTM-Mixer effectively captures cross-variate interactions.

**Ensembling over More than Two Views.** Following the perspective that multi-view mixing is an ensemble over variate orderings, we extend our method to accommodate additional randomly permuted views beyond the original and reverse order. To this end, $\mathrm{FC}^{\mathrm{view}}$ is extended accordingly. The full results in App. I show that this does not improve the modeling accuracy of xLSTM-Mixer.

Table 3: **Each component of xLSTM-Mixer is essential for its overall strong performance.** The notation follows Tab. 1. Results are averages over three seeds.

| | | #1 (full) | #2 | #3 | #4 | #5 | #6 | #7 | #8 | #9 | #10 | #11 | #12 | #13 | #14 |
|---|---|---|---|---|---|---|---|---|---|---|---|---|---|---|---|
| | Time Mixing | ✓ | ✓ | ✓ | ✓ | ✓ | DLinear | ✓ | ✓ | ✓ | ✓ | ✗ | ✗ | ✗ | ✗ |
| | xLSTM type | sLSTM | mLSTM | LSTM | GRU | sLSTM | sLSTM | sLSTM | sLSTM | sLSTM | None | sLSTM | sLSTM | sLSTM | sLSTM |
| | Recurr. order | Variates | Variates | Variates | Variates | Time | Variates | Variates | Variates | Variates | None | Variates | Variates | Variates | Variates |
| | Init. Token | ✓ | ✓ | ✓ | ✓ | ✓ | ✓ | ✗ | ✓ | ✗ | ✗ | ✓ | ✗ | ✓ | ✗ |
| | View Mixing | ✓ | ✓ | ✓ | ✓ | ✓ | ✓ | ✓ | ✗ | ✗ | ✗ | ✓ | ✓ | ✗ | ✗ |
| | Metric | MSE MAE | MSE MAE | MSE MAE | MSE MAE | MSE MAE | MSE MAE | MSE MAE | MSE MAE | MSE MAE | MSE MAE | MSE MAE | MSE MAE | MSE MAE | MSE MAE |
| Weather | 96 | **0.143 0.184** | 0.148 0.192 | 0.148 0.190 | 0.147 0.193 | 0.148 0.194 | 0.145 0.187 | 0.145 0.186 | 0.144 0.185 | 0.144 0.186 | 0.173 0.223 | 0.149 0.193 | 0.151 0.195 | 0.149 0.192 | 0.152 0.195 |
| | 192 | **0.186 0.226** | 0.193 0.235 | 0.191 0.232 | 0.188 0.233 | 0.196 0.239 | 0.188 0.229 | 0.188 0.228 | **0.186 0.226** | 0.188 0.228 | 0.219 0.257 | 0.192 0.233 | 0.192 0.234 | 0.191 0.234 | 0.193 0.236 |
| | 336 | **0.237 0.266** | 0.241 0.272 | 0.241 0.271 | 0.238 0.272 | 0.252 0.281 | **0.237** 0.267 | 0.239 0.267 | 0.241 0.270 | 0.242 0.270 | 0.261 0.288 | 0.240 0.271 | 0.242 0.273 | 0.242 0.273 | 0.244 0.274 |
| | 720 | 0.310 0.324 | 0.313 0.325 | 0.327 0.344 | 0.345 0.361 | 0.315 0.328 | 0.312 0.325 | 0.310 0.324 | **0.309 0.323** | **0.309 0.323** | 0.320 0.334 | 0.320 0.329 | 0.319 0.329 | 0.322 0.330 | 0.319 0.328 |
| ETTm1 | 96 | 0.275 0.328 | 0.285 0.339 | 0.317 0.367 | 0.312 0.361 | 0.298 0.348 | **0.274** 0.329 | 0.277 0.329 | 0.278 0.331 | 0.279 0.333 | 0.295 0.338 | 0.282 0.339 | 0.285 0.341 | 0.281 0.337 | 0.284 0.339 |
| | 192 | 0.319 0.354 | 0.329 0.365 | 0.354 0.388 | 0.349 0.382 | 0.337 0.369 | **0.319** 0.356 | 0.321 **0.354** | 0.321 0.356 | 0.322 0.358 | 0.329 0.357 | 0.329 0.364 | 0.330 0.365 | 0.337 0.367 | 0.335 0.366 |
| | 336 | 0.353 **0.374** | 0.363 0.384 | 0.383 0.403 | 0.375 0.395 | 0.368 0.388 | **0.351** 0.376 | 0.354 0.375 | 0.355 0.377 | 0.357 0.379 | 0.359 0.376 | 0.367 0.385 | 0.367 0.385 | 0.366 0.384 | 0.366 0.385 |
| | 720 | **0.409 0.407** | 0.417 0.414 | 0.440 0.432 | 0.447 0.441 | 0.420 0.416 | **0.409** 0.408 | 0.411 0.408 | 0.413 0.411 | 0.414 0.411 | 0.412 **0.407** | 0.422 0.412 | 0.422 0.413 | 0.417 0.410 | 0.418 0.411 |

## 4.5 Ablation Studies

To assess the contributions of each component in xLSTM-Mixer to its strong forecast performance, we conducted ablation studies of thirteen different model configurations with the results listed in Tab. 3. Each configuration represents a different combination of the four key components: mixing time with NLinear/DLinear, using sLSTM/mLSTM/LSTM/GRU for joint mixing, striding over the variate or time dimension, learning initial embedding tokens $\eta$, and multi-view mixing.

The full version of xLSTM-Mixer (#1), which integrates all components, achieves the best performance overall. Ablating components of xLSTM-Mixer each causes both error metrics to increase, entailing that they contribute positively to the overall architecture. Specifically, omitting xLSTM (only having LSTM, #3) raises the MAE by 6.2%/the MSE by 7.0%, variate recurrence (#5) by 4.3%/4.7%, learning initial token embedding (#7) by 0.4%/0.7%, view mixing (#8) by 0.6%/0.7%, and time mixing (#11) by 2.7%/3.1%. For example, removing the time mixing (#11) increases the MAE by 3.4% on ETTm1 at length 96 or 3.1% at length 192, highlighting its critical role in capturing cross-time dependencies. When we now omit everything except for time mixing on Weather at 192, we suffer a 13.7% performance decrease. Additionally, substituting sLSTM blocks with mLSTM, LSTM, or GRU (#2–#4) consistently degrades performance, particularly for LSTM and GRU at longer horizons. We attribute this to the sLSTM's inherent structure, which provides stronger mixing capabilities and mitigates the degradation commonly observed over extended prediction windows. However, we also observe that some configurations of xLSTM-Mixer, which exclude specific components, remain competitive. For instance, #7, which excludes the initial embedding token, still performs very well. This suggests that while it contributes positively to the overall performance, the model can sometimes still achieve competitive results without it. Similarly, depending on the dataset and target metric, initial forecasting with DLinear instead of NLinear is a viable option, too (#6).

The ablation study confirms that all components of xLSTM-Mixer contribute to its effectiveness, with the full configuration yielding the best results. Furthermore, we identified the sLSTM blocks and time-mixing components as critical for ensuring high accuracy across datasets and prediction lengths.

## 5 Related Work

**Time Series Forecasting.** A long line of machine learning research led from early statistical methods like ARIMA [Sims, 1980, Box et al., 2015] to contemporary models based on deep learning, where six architectural families take center stage: based on recurrence, state spaces, convolutions, Multilayer Perceptrons (MLPs), mixing, and Transformers. While all of them are used by practitioners today, the research focus for long-term time series forecasting is gradually shifting over time. Initially, the naturally sequential recurrent models such as Long Short-Term Memory (LSTM) [Hochreiter and Schmidhuber, 1997] and Gated Recurrent Units (GRUs) [Cho et al., 2014] were used for time series analysis. Their main benefits are the high inference efficiency and arbitrary input and output lengths due to their autoregressive nature. While their effectiveness has historically been constrained by a limited ability to capture long-range dependencies, active research remains to alleviate these limitations [Salinas et al., 2020], including the xLSTM architecture presented in Sec. 2 [Beck et al.,

2024, Alharthi and Mahmood, 2024] and SutraNets [Bergsma et al., 2023]. Closely related are state space models (SSMs) such as Mamba [Gu and Dao, 2024, Wang et al., 2025b] or Chimera [Behrouz et al., 2024], which permit parallel inference for improved efficiency. Similarly efficient, yet more restricted in their output length, are the location-invariant CNNs [Li et al., 2022, Lara-Benítez et al., 2021], such as TCN [Lea et al., 2016], TimesNet [Wu et al., 2022a], and MICN [Wang et al., 2022]. Recently, some MLP-based architectures have also shown good success, including the simplistic DLinear and NLinear models [Zeng et al., 2023], the encoder-decoder architecture of TiDE [Das et al., 2023], and the older hierarchical N-BEATS [Oreshkin et al., 2019] and N-HiTS [Challu et al., 2023] models. They are closely related to other mixing architectures such as TimeMixer(++) [Wang et al., 2024a, 2025a] and TSMixer [Chen et al., 2023c]. A lot of accurate models with significant compute costs have been proposed based on Transformers [Vaswani et al., 2017], such as Autoformer [Wu et al., 2021], TFT [Lim et al., 2021], FEDFormer [Zhou et al., 2022], PatchTST [Nie et al., 2023], and iTransformer [Liu et al., 2023]. Finally, the most recent development are pretrained models fitted on multiple time series datasets [Kraus et al., 2024]. Examples include Chronos [Ansari et al., 2024], Moirai [Woo et al., 2024], and Timer-XL [Liu et al., 2025]. xLSTM-Mixer combines two model families, namely the highly expressive mixing models with efficient recurrence, to benefit from the strengths of both.

**xLSTM Models for Time Series.**   Some initial experiments of applying xLSTMs [Beck et al., 2024] to time series were already performed by Alharthi and Mahmood [2024] with their proposed xLSTMTime model. While it showed promising forecasting performance, these initial soundings did not surpass stronger models at that time, such as TimeMixer [Wang et al., 2024a], on multivariate benchmarks. Furthermore, despite our best efforts, the experimental findings unfortunately could not be replicated with either the official implementation or the scarce details in the paper. We ensure that our method xLSTM-Mixer is well-suited as a foundation for further research by providing extensive model analysis, an ablation study with 13 variations, and ensuring that results are readily reproducible. Our methodology draws from xLSTMTime yet improves on it by several key components. Most importantly, our novel multi-view mixing consistently enhances forecasting performance. Furthermore, we find the trend-seasonality decomposition redundant and a simple NLinear normalization scheme [Zeng et al., 2023] to suffice. Concurrently, Kong et al. [2024] also investigate using xLSTMs for time series forecasting, arriving at similar conclusions. Poonia et al. [2025] successfully employ xLSTMs to detect Granger Causality in time series.

## 6   Conclusion

In this work, we introduced xLSTM-Mixer, a method that combines a linear forecast with refinement through xLSTM blocks. Our architecture integrates time, joint, and view mixing to capture complex dependencies. In long-term forecasting, xLSTM-Mixer consistently achieves state-of-the-art performance, outperforming a large set of previous methods in 27 out of 56 cases. We also evaluate xLSTM-Mixer in heterogeneous probabilistic settings on GIFT-Eval, where it delivers strong, competitive performance. Beyond forecasting, xLSTM-Mixer performs strongly on time-series classification when used as an embedding model. Moreover, xLSTM-Mixer attains these results at a low memory footprint. Our model analysis provided insights into the contribution of each component and demonstrated robustness to hyperparameter variations.

**Limitations.**   While xLSTM-Mixer achieves strong accuracy with limited compute, certain assumptions need to be met for it to be applicable. Specifically, it assumes that all variables are sampled on a uniform time grid, meaning that irregular or missing timestamps must still be handled in pre-processing. Furthermore, treating variates as the sequential axis ties runtime and memory directly to the number of channels, which can become a bottleneck for extremely high-dimensional multivariate time-series. Moreover, the simultaneous mixing of multiple views blends temporal and cross-channel information in ways that make detailed attributions difficult.

**Future Work.**   Addressing these aspects through adaptive variate grouping, continuous-time embeddings, and lightweight explanation modules are potential paths forward beyond the current work. Finally, extending xLSTM-Mixer to tasks such as imputation or anomaly detection offers promising future directions.

## Acknowledgements

This work received funding from the EU project EXPLAIN, under the German Federal Ministry of Research, Technology and Space (BMFTR) (grant 16IS22030D). Furthermore, it was funded by the ACATIS Investment KVG mbH project "Temporal Machine Learning for Long-Term Value Investing" and the BMFTR KompAKI project within the "The Future of Value Creation – Research on Production, Services and Work" program (funding number 02L19C150), managed by the Project Management Agency Karlsruhe (PTKA). The author of Eindhoven University of Technology received support from their Department of Mathematics and Computer Science and the Eindhoven Artificial Intelligence Systems Institute. Furthermore, this work benefited from the HMWK project "The Third Wave of Artificial Intelligence - 3AI", the BMFTR project "XEI" (FKZ 16IS24079B), and from early stages of the Clusters of Excellence "Reasonable AI" (EXC-3057) and "The Adaptive Mind" (EXC-3066) funded by the German Research Foundation (DFG) under Germany's Excellence Strategy; funding will begin in 2026. The authors are responsible for the content of this publication.

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

# A   Societal Impact

This paper presents work whose goal is to advance the field of Machine Learning. There are many potential societal consequences of our work, none which we feel must be specifically highlighted here.

Our research advances machine learning by enhancing the capabilities of long-term forecasting in time series models, significantly improving both accuracy and efficiency. By developing xLSTM-Mixer, we introduce a robust framework that can be applied across various industries, including finance, healthcare, energy, and logistics. The improved forecasting accuracy enables better decision-making in critical areas, such as optimizing resource allocation, predicting market trends, and managing risk.

However, we also recognize the potential risks associated with the misuse of these advanced models. Time series forecasting models could be leveraged for malicious purposes, especially when applied at scale. For example, in the financial sector, adversarial agents might manipulate forecasts to create market instability. In political or social contexts, these models could be exploited to predict and influence public opinion or destabilize economies. Additionally, the application of these models in sensitive domains like healthcare and security may lead to unintended consequences if not carefully regulated and ethically deployed.

Therefore, it is essential that the use of xLSTM-Mixer, like all machine learning technologies, is guided by responsible practices and ethical considerations. We encourage stakeholders to adopt rigorous evaluation processes to ensure fairness, transparency, and accountability in its deployment, and to remain vigilant to the broader societal implications of time series forecasting technologies.

# B   Rationale for Employing sLSTM over mLSTM

Beck et al. [2024] present a comprehensive case for xLSTM blocks over earlier recurrent models. Here, we explain why, within those, we choose scalar-memory sLSTM blocks as our mixing primitive rather than mLSTM. The primary motivation is the state tracking via memory mixing. sLSTM preserves hidden-to-hidden recurrences in its gates and cell update, enabling conditional, history-aware updates. mLSTM removes these paths to allow full parallelization, which limits this state tracking capability.

**Mechanisms of sLSTM and mLSTM.**   The sLSTM block maintains the LSTM-style memory mixing, augmented with exponential gates and a stabilization term:

$$c_t = f_t c_{t-1} + i_t z_t, \qquad h_t = o_t g(c_t). \tag{9}$$

Here, $f_t, i_t, o_t$ each depend on $h_{t-1}$ through recurrent matrices (Beck et al. [2024], Greff et al. [2017]), allowing the block to blend past and present conditionally. By contrast, mLSTM writes key-value pairs into a matrix memory:

$$C_t = f_t C_{t-1} + i_t v_t k_t^\top, \tag{10}$$

with gates that depend only on the current input (no hidden-to-hidden paths) [Beck et al., 2024, Sec. 2.3]. This design eliminates memory mixing. Note that we use the scalar formulation for simplicity, whereas Sec. 2.1 directly presented the full vector-valued xLSTMs.

**Importance of Latent Phases.**   In long-horizon forecasting, tracking latent phases (e.g., heating $\leftrightarrow$ cooling; up $\leftrightarrow$ down) is critical. With sLSTM, gates can be trained such that, upon detection of a switch cue, the cell overwrites ($i_t \approx 1, f_t \approx 0$); otherwise, it retains ($i_t \approx 0, f_t \approx 1$). Because the gates read $h_{t-1}$, the decision is conditional on the current internal state. In mLSTM, gates cannot use $h_{t-1}$, meaning updates become fixed linear functionals of the input history. Empirically, architectures lacking memory mixing (mLSTM, Mamba, Transformers) fail simple state-tracking tasks such as parity, as shown by Merrill et al. [2024] and Beck et al. [2024].

**Design Consequences.**   For long-horizon forecasting, state-tracking capacity outweighs raw matrix-memory capacity, precluding the direct learning of inter-token relations during joint mixing. We, thus, exclusively employ sLSTM in xLSTM-Mixer. In our ablations, replacing sLSTM with mLSTM, GRU, or LSTM consistently harms long-horizon accuracy, particularly at 336/720 steps (cf. Tab. 3). Prior and concurrent work supports this finding: xLSTMTime uses sLSTM on smaller forecasting datasets where precise phase tracking dominates and opts for mLSTM only when raw storage capacity is paramount [Alharthi and Mahmood, 2024]. TiRex likewise attributes gains to retaining state tracking [Auer et al., 2025].

# C  Implementation Details

**Experimental Details.**   Our codebase is implemented in Python 3.11, leveraging *PyTorch* version 2. [Paszke et al., 2019] in combination with *Lightning* version 2.4[2] for model training and optimization. We used the custom CUDA implementation[3] for sLSTM, which relies on NVIDIA Compute Capability $\geq 8.0$. Thus, our experiments were conducted on a single NVIDIA A100 80GB GPU. The majority of our baseline implementations, along with data loading and preprocessing steps, are adapted from the *Time-Series-Library*[4] of Wang et al. [2024b]. For GIFT-Eval [Aksu et al., 2024], we integrated xLSTM-Mixer into the official evaluation harness via a *GluonTS*-style [Alexandrov et al., 2020] estimator. For xLSTMTime, we used code based on the official repository[5]. We employ *Captum*[6] [Kokhlikyan et al., 2020] to compute the SHAP values used in model analysis. We used *scikit-posthocs* [Terpilowski, 2019] for significance analyses.

**Training and Hyperparameters.**   We optimized xLSTM-Mixer in 32 bits for up to 60 epochs with a cosine-annealing scheduler with the Adam optimizer [Kingma and Ba, 2015], using $\beta_1 = 0.9$ and $\beta_2 = 0.999$ and no weight decay. Hyperparameter (HP) tuning was conducted using Optuna [Akiba et al., 2019] with the choices provided in Tab. 4. We optimized for the L1 forecast error, also known as the Mean Absolute Error (MAE). To further stabilize the training process, gradient clipping with a maximum norm of 1.0 was applied. All experiments were run with the three different random seeds $\{2021, 2022, 2023\}$. For most models, the initial publications already provide HPs appropriate for the well-known datasets, where we thus directly adopt these results. For xLSTMTime, this was not the case, and we were not able to fully reproduce the results in the paper despite our best efforts (cf. Sec. 5). We still present the better results from Alharthi and Mahmood [2024] so as not to erroneously underestimate the method. For the datasets ETTh2, ETTm1, and ETTm2 without such results, we ran individual HP searches analogously to our method to ensure a fair comparison.

Table 4: **Hyperparameters and their choices.**

| Hyperparameter | Choices |
| --- | --- |
| Batch size | $\{16, 32, 64, 128, 256, 512\}$ |
| Initial learning rate | $\{1 \cdot 10^{-2}, 3 \cdot 10^{-3}, 1 \cdot 10^{-3}, 5 \cdot 10^{-4}, 2 \cdot 10^{-4}, 1 \cdot 10^{-4}\}$ |
| Scheduler warmup steps | $\{5, 10, 15\}$ |
| Lookback length | $\{96, 256, 512, 768, 1024, 2048\}$ |
| Embedding dimension $D$ | $\{32, 64, 128, 256, 512, 768, 1024\}$ |
| sLSTM conv. kernel width | $\{$disabled, 2, 4$\}$ |
| sLSTM dropout rate | $\{0.1, 0.25\}$ |
| # sLSTM blocks $M$ | $\{1, 2, 3, 4\}$ |
| # sLSTM heads | $\{4, 8, 16, 32\}$ |

**Forecasting Metrics.**   We follow common practice in the literature [Wu et al., 2021, Wang et al., 2024a] for maximum comparability and, therefore, evaluate deterministic long-term forecasting of all models on the mean absolute error (MAE), mean squared error (MSE). For probabilistic evaluation (as in GIFT-Eval [Aksu et al., 2024]), we use the Mean Absolute Scaled Error (MASE) and the Continuous Ranked Probability Score (CRPS). All metrics are computed per variate and averaged over all variates. Formally,

$$\text{MAE}(\boldsymbol{y}, \hat{\boldsymbol{y}}) = \sum_{i=1}^{H} |y_i - \hat{y}_i|, \qquad \text{MSE}(\boldsymbol{y}, \hat{\boldsymbol{y}}) = \sum_{i=1}^{H} (y_i - \hat{y}_i)^2,$$

$$\text{MASE}(\boldsymbol{y}, \hat{\boldsymbol{y}}) = \frac{1}{H} \sum_{t=1}^{H} \frac{|y_t - \hat{y}_t|}{d}, \qquad d = \frac{1}{T-m} \sum_{t=m+1}^{T} |y_t - y_{t-m}|.$$

---

[2] https://lightning.ai/pytorch-lightning

[3] https://github.com/NX-AI/xlstm

[4] https://github.com/thuml/Time-Series-Library

[5] https://github.com/muslehal/xLSTMTime

[6] https://captum.ai

The CRPS for predictive CDF $\hat{F}$ and outcome $y$ is

$$\text{CRPS}(\hat{F}, y) = \int_{-\infty}^{\infty} \left(\hat{F}(z) - \mathbb{1}\{y \leq z\}\right)^2 \mathrm{d}z \; = \; \mathbb{E}|X - y| \; - \; \tfrac{1}{2}\mathbb{E}|X - X'|,$$

where $\boldsymbol{y}$ are the targets, $\hat{\boldsymbol{y}}$ the predictions; $H$ is the horizon length, $T$ the in-sample length, $m$ a seasonal period, $d$ the MASE scaling constant (computed per variate), $\hat{F}$ the predictive CDF, and $X, X' \sim \hat{F}$ i.i.d. When forecasts are provided as quantiles $\{\hat{q}_{t,\tau_k}\}_{k=1}^{K}$, we use the standard discrete approximation to CRPS via the pinball loss:

$$\widehat{\text{CRPS}}_t = \frac{2}{K}\sum_{k=1}^{K} \rho_{\tau_k}(y_t - \hat{q}_{t,\tau_k}), \qquad \rho_{\tau}(u) = (\tau - \mathbb{1}\{u < 0\})\, u,$$

$$\overline{\text{CRPS}} = \frac{1}{H}\sum_{t=1}^{H} \widehat{\text{CRPS}}_t.$$

## D   Benchmark Datasets

Tab. 5 provides an overview of the datasets we used to compare xLSTM-Mixer with other time series forecasting models. The last column shows the range of Hurst exponents [Hurst, 1951] over the variates measuring long-term patterns. The larger the values are over 0.5, the more long-term patterns are in the time series.

Table 5: **The long-term forecasting benchmark datasets and their key properties.**

| Dataset | Source | Domain | Horizons | Sampling | #Variates | Hurst exp. |
|---|---|---|---|---|---|---|
| Weather | Zhou et al. [2021] | Weather | 96–720 | 10 min | 21 | 0.333–1.000 |
| Electricity | Zhou et al. [2021] | Power Usage | 96–720 | 1 hour | 321 | 0.555–1.000 |
| Traffic | Wu et al. [2021] | Traffic Load | 96–720 | 1 hour | 862 | 0.162–1.000 |
| ETT | Zhou et al. [2021] | Power Production | 96–720 | 15&60 min | 7 | 0.906–1.000 |

## E   Full Results for Long-Term Forecasting

Tab. 6 shows the full results for long-term forecasting. This work involves conducting all experiments three times using seeds 2021, 2022, and 2023, following the setup of prior research [Wu et al., 2021, Nie et al., 2023, Wang et al., 2024a]. We therefore present the standard deviation of our model and the second-best models in terms of MSE and MAE in Tab. 7. This table, along with our experiments described in Fig. 4a and Fig. 4b, further underscores the robustness of xLSTM-Mixer.

Table 7: **xLSTM-Mixer provides state-of-the-art performance at low variance across datasets.** This table shows the average performance and average standard deviation over all four prediction lengths in Tab. 6. They are contextualized by the competitive baselines TimeMixer and TiDE.

| Model | xLSTM-Mixer (ours) | | TimeMixer | | TiDE | |
|---|---|---|---|---|---|---|
| Metric | MSE | MAE | MSE | MAE | MSE | MAE |
| Weather | 0.219 ±0.000 | 0.250 ±0.000 | 0.240 ±0.010 | 0.271 ±0.009 | 0.236 ±0.001 | 0.282 ±0.001 |
| Electricity | 0.153 ±0.001 | 0.245 ±0.001 | 0.182 ±0.017 | 0.272 ±0.006 | 0.159 ±0.002 | 0.257 ±0.001 |
| Traffic | 0.392 ±0.000 | 0.253 ±0.000 | 0.484 ±0.015 | 0.297 ±0.013 | 0.356 ±0.001 | 0.261 ±0.001 |
| ETTh1 | 0.397 ±0.001 | 0.420 ±0.001 | 0.047 ±0.002 | 0.440 ±0.005 | 0.419 ±0.000 | 0.430 ±0.000 |
| ETTh2 | 0.340 ±0.001 | 0.382 ±0.000 | 0.364 ±0.008 | 0.375 ±0.010 | 0.345 ±0.002 | 0.394 ±0.001 |
| ETTm1 | 0.339 ±0.000 | 0.366 ±0.000 | 0.381 ±0.003 | 0.395 ±0.006 | 0.355 ±0.000 | 0.378 ±0.000 |
| ETTm2 | 0.248 ±0.001 | 0.307 ±0.001 | 0.275 ±0.001 | 0.323 ±0.003 | 0.249 ±0.000 | 0.312 ±0.000 |

Table 6: **Full long-term forecasting results for Tab. 1.** *Avg* is averaged from all four prediction lengths {96, 192, 336, 720}. A lower MSE or MAE indicates a better prediction. The **best** result for each dataset is highlighted bold red and second-best blue and underlined. Wins for each model out of all 28 settings are shown at the bottom.

| Models | | Recurrent | | | Mixer | | | MLP | | | State Space | | Transformer | | Convolutional | | Pretrained[*] | |
|---|---|---|---|---|---|---|---|---|---|---|---|---|---|---|---|---|---|---|
| | | xLSTM-Mixer | xLSTMTime 2024 | LSTM 1997[†] | TimeMix.++ 2025a | TimeMix. 2024a | TSMixer 2023c | CycleNet 2024 | DLinear 2023 | TiDE 2023 | S-Mamba 2025b | Chimera 2024 | PatchTST 2023 | iTransf. 2023 | Mod.TCN 2023 | TimesNet 2022a | Timer-XL 2025 | Moirai_Base 2024 |
| | Metric | MSE MAE | MSE MAE | MSE MAE | MSE MAE | MSE MAE | MSE MAE | MSE MAE | MSE MAE | MSE MAE | MSE MAE | MSE MAE | MSE MAE | MSE MAE | MSE MAE | MSE MAE | MSE MAE | MSE MAE |
| Weather | 96 | **0.143 0.184** | 0.144 0.187 | 0.369 0.406 | 0.155 0.205 | 0.147 0.197 | 0.145 0.198 | 0.148 0.200 | 0.176 0.237 | 0.166 0.222 | 0.165 0.210 | 0.146 0.206 | 0.149 0.198 | 0.174 0.214 | 0.149 0.200 | 0.172 0.220 | 0.171 0.225 | 0.220 0.217 |
| | 192 | **0.186 0.226** | 0.192 0.236 | 0.416 0.435 | 0.201 0.245 | 0.189 0.239 | 0.191 0.242 | 0.190 0.240 | 0.220 0.282 | 0.209 0.263 | 0.214 0.252 | 0.189 0.239 | 0.194 0.241 | 0.221 0.254 | 0.196 0.245 | 0.219 0.261 | 0.221 0.271 | 0.271 0.259 |
| | 336 | **0.236 0.266** | 0.237 0.272 | 0.455 0.454 | 0.237 0.265 | 0.241 0.280 | 0.242 0.280 | 0.242 0.283 | 0.265 0.319 | 0.254 0.301 | 0.274 0.297 | 0.244 0.281 | 0.306 0.282 | 0.278 0.296 | 0.238 0.277 | 0.280 0.306 | 0.274 0.311 | 0.286 0.297 |
| | 720 | 0.310 0.323 | 0.313 0.326 | 0.535 0.520 | 0.312 0.334 | 0.310 0.330 | 0.320 0.336 | 0.312 0.333 | 0.323 0.362 | 0.313 0.340 | 0.350 0.345 | **0.297 0.309** | 0.314 0.334 | 0.358 0.347 | 0.314 0.334 | 0.365 0.359 | 0.356 0.370 | 0.373 0.354 |
| | Avg | **0.219 0.250** | 0.222 0.255 | 0.444 0.454 | 0.226 0.262 | 0.222 0.262 | 0.225 0.264 | 0.223 0.264 | 0.246 0.300 | 0.236 0.282 | 0.251 0.276 | **0.219** 0.258 | 0.241 0.264 | 0.258 0.278 | 0.224 0.264 | 0.259 0.287 | 0.256 0.294 | 0.287 0.281 |
| Electricity | 96 | **0.126 0.218** | 0.128 0.221 | 0.375 0.437 | 0.135 0.222 | 0.129 0.224 | 0.131 0.229 | **0.126** 0.221 | 0.140 0.237 | 0.132 0.229 | 0.139 0.235 | 0.132 0.234 | 0.129 0.222 | 0.148 0.240 | 0.129 0.226 | 0.168 0.272 | 0.141 0.237 | 0.160 0.250 |
| | 192 | 0.144 0.235 | 0.150 0.243 | 0.442 0.473 | 0.147 0.235 | **0.140 0.220** | 0.151 0.246 | 0.144 0.237 | 0.153 0.249 | 0.147 0.243 | 0.159 0.255 | 0.144 0.223 | 0.147 0.240 | 0.162 0.253 | 0.143 0.239 | 0.184 0.289 | 0.159 0.254 | 0.175 0.263 |
| | 336 | 0.157 **0.250** | 0.166 0.259 | 0.439 0.473 | 0.164 0.245 | 0.161 0.255 | 0.161 0.261 | 0.159 0.255 | 0.169 0.267 | 0.161 0.261 | 0.176 0.272 | **0.156** 0.259 | 0.163 0.259 | 0.178 0.269 | 0.161 0.259 | 0.198 0.300 | 0.177 0.272 | 0.187 0.277 |
| | 720 | **0.183 0.276** | 0.185 0.276 | 0.980 0.814 | 0.212 0.310 | 0.194 0.287 | 0.197 0.293 | 0.196 0.290 | 0.203 0.301 | 0.196 0.294 | 0.204 0.298 | 0.184 0.280 | 0.197 0.290 | 0.225 0.317 | 0.191 0.286 | 0.220 0.320 | 0.219 0.308 | 0.228 0.309 |
| | Avg | **0.153 0.245** | 0.157 0.250 | 0.559 0.549 | 0.165 0.253 | 0.156 0.246 | 0.160 0.256 | 0.156 0.251 | 0.166 0.264 | 0.159 0.267 | 0.170 0.265 | 0.154 0.249 | 0.159 0.253 | 0.178 0.270 | 0.156 0.253 | 0.192 0.295 | 0.174 0.278 | 0.187 0.274 |
| Traffic | 96 | 0.357 **0.236** | 0.358 0.242 | 0.843 0.453 | 0.392 0.253 | 0.360 0.249 | 0.376 0.264 | 0.374 0.268 | 0.410 0.282 | **0.336** 0.253 | 0.382 0.261 | 0.366 0.248 | 0.360 0.249 | 0.395 0.268 | 0.368 0.253 | 0.593 0.321 | –[‡] – | –[‡] – |
| | 192 | 0.377 **0.241** | 0.378 0.253 | 0.847 0.453 | 0.402 0.258 | 0.375 0.250 | 0.397 0.277 | 0.390 0.275 | 0.423 0.287 | **0.346** 0.257 | 0.396 0.267 | 0.394 0.292 | 0.379 0.256 | 0.417 0.276 | 0.379 0.261 | 0.617 0.336 | – – | – – |
| | 336 | 0.394 **0.250** | 0.392 0.261 | 0.853 0.455 | 0.428 0.263 | 0.385 0.270 | 0.413 0.290 | 0.405 0.282 | 0.436 0.296 | **0.355 0.260** | 0.417 0.276 | 0.409 0.311 | 0.392 0.264 | 0.433 0.283 | 0.397 0.270 | 0.629 0.336 | – – | – – |
| | 720 | 0.439 0.283 | 0.434 0.287 | 1.500 0.805 | 0.441 0.282 | 0.430 0.281 | 0.444 0.306 | 0.441 0.302 | 0.466 0.315 | **0.386 0.273** | 0.460 0.300 | 0.443 0.294 | 0.432 0.286 | 0.467 0.302 | 0.440 0.296 | 0.640 0.350 | – – | – – |
| | Avg | 0.392 **0.253** | 0.391 0.261 | 1.011 0.541 | 0.416 0.264 | 0.387 0.262 | 0.408 0.284 | 0.403 0.282 | 0.434 0.295 | **0.356** 0.261 | 0.414 0.276 | 0.403 0.286 | 0.391 0.264 | 0.428 0.282 | 0.396 0.270 | 0.620 0.336 | – – | – – |
| ETTh1 | 96 | **0.359 0.386** | 0.368 0.395 | 1.044 0.773 | 0.361 0.403 | 0.361 0.390 | 0.361 0.392 | 0.382 0.403 | 0.375 0.399 | 0.375 0.398 | 0.386 0.405 | 0.362 0.391 | 0.370 0.400 | 0.386 0.405 | 0.368 0.394 | 0.384 0.402 | 0.369 0.391 | 0.376 0.392 |
| | 192 | 0.402 0.417 | 0.401 0.416 | 1.217 0.832 | 0.416 0.441 | 0.409 0.414 | 0.404 0.418 | 0.421 0.426 | 0.405 0.416 | 0.412 0.422 | 0.443 0.437 | **0.398 0.415** | 0.413 0.429 | 0.441 0.436 | 0.405 0.413 | 0.412 0.413 | 0.405 0.413 | 0.433 0.428 |
| | 336 | 0.408 0.429 | 0.422 0.437 | 1.259 0.841 | 0.430 0.434 | 0.430 0.429 | 0.420 0.431 | 0.449 0.444 | 0.439 0.443 | 0.435 0.433 | 0.489 0.468 | 0.402 0.416 | 0.422 0.440 | 0.487 0.458 | **0.391 0.412** | 0.491 0.469 | 0.418 0.423 | 0.433 0.428 |
| | 720 | **0.419** 0.448 | 0.441 0.465 | 1.271 0.838 | 0.467 0.451 | 0.445 0.460 | 0.463 0.472 | 0.486 0.487 | 0.472 0.490 | 0.454 0.465 | 0.502 0.489 | 0.458 0.477 | 0.447 0.468 | 0.503 0.491 | 0.450 0.461 | 0.521 0.500 | 0.423 **0.441** | 0.447 0.444 |
| | Avg | **0.397 0.420** | 0.408 0.428 | 1.198 0.821 | 0.419 0.432 | 0.411 0.423 | 0.412 0.428 | 0.435 0.440 | 0.423 0.437 | 0.419 0.430 | 0.455 0.450 | 0.405 0.424 | 0.413 0.434 | 0.454 0.448 | 0.404 0.420 | 0.458 0.450 | 0.404 **0.417** | 0.417 0.419 |
| ETTh2 | 96 | 0.267 0.329 | 0.273 0.333 | 2.522 1.278 | 0.276 0.328 | 0.271 0.330 | 0.274 0.341 | 0.293 0.352 | 0.289 0.353 | 0.270 0.336 | 0.296 0.348 | **0.257 0.325** | 0.274 0.337 | 0.297 0.349 | 0.263 0.332 | 0.340 0.374 | 0.283 0.342 | 0.294 0.330 |
| | 192 | 0.338 0.375 | 0.340 0.378 | 3.312 1.384 | 0.342 0.379 | 0.317 0.402 | 0.339 0.385 | 0.359 0.395 | 0.383 0.418 | 0.332 0.380 | 0.376 0.396 | **0.314 0.369** | 0.314 0.382 | 0.380 0.400 | 0.320 0.374 | 0.402 0.414 | 0.340 0.379 | 0.365 0.375 |
| | 336 | 0.367 0.401 | 0.373 0.403 | 3.291 1.388 | 0.344 0.398 | 0.332 0.396 | 0.361 0.406 | 0.392 0.423 | 0.448 0.465 | 0.360 0.407 | 0.424 0.446 | **0.316 0.381** | 0.329 0.384 | 0.428 0.432 | 0.366 0.400 | 0.452 0.452 | 0.366 0.408 | 0.416 0.433 |
| | 720 | 0.388 0.424 | 0.398 0.430 | 3.257 1.357 | 0.392 **0.415** | 0.342 0.408 | 0.445 0.470 | 0.425 0.451 | 0.605 0.551 | 0.419 0.451 | 0.426 0.444 | 0.388 0.427 | 0.379 0.422 | 0.427 0.445 | 0.392 0.433 | 0.462 0.468 | 0.397 0.431 | 0.416 0.433 |
| | Avg | 0.340 0.382 | 0.346 0.386 | 3.095 1.352 | 0.339 0.380 | **0.316** 0.384 | 0.355 0.401 | 0.367 0.405 | 0.431 0.447 | 0.345 0.394 | 0.381 0.405 | 0.318 **0.375** | 0.324 0.381 | 0.383 0.407 | 0.322 0.379 | 0.414 0.427 | 0.347 0.388 | 0.362 0.382 |
| ETTm1 | 96 | **0.275 0.328** | 0.286 0.335 | 0.863 0.664 | 0.310 0.334 | 0.291 0.340 | 0.285 0.339 | 0.297 0.351 | 0.299 0.343 | 0.306 0.349 | 0.333 0.368 | 0.293 0.351 | 0.293 0.346 | 0.334 0.368 | 0.292 0.346 | 0.338 0.375 | 0.317 0.356 | 0.363 0.356 |
| | 192 | **0.319 0.354** | 0.329 0.361 | 1.113 0.776 | 0.348 0.362 | 0.327 0.365 | 0.327 0.365 | 0.338 0.377 | 0.335 0.365 | 0.335 0.366 | 0.376 0.390 | 0.329 0.362 | 0.333 0.370 | 0.377 0.391 | 0.332 0.368 | 0.374 0.387 | 0.358 0.381 | 0.388 0.375 |
| | 336 | 0.353 0.374 | 0.358 0.379 | 1.267 0.832 | 0.376 0.391 | 0.360 0.381 | 0.356 0.382 | 0.374 0.400 | 0.369 0.386 | 0.364 0.384 | 0.408 0.413 | **0.352 0.383** | 0.369 0.392 | 0.426 0.420 | 0.365 0.391 | 0.410 0.411 | 0.386 0.401 | 0.416 0.392 |
| | 720 | **0.409 0.407** | 0.416 0.411 | 1.324 0.858 | 0.440 0.423 | 0.415 0.417 | 0.419 0.414 | 0.431 0.425 | 0.425 0.421 | 0.413 0.413 | 0.475 0.448 | **0.408** 0.412 | 0.416 0.420 | 0.491 0.459 | 0.416 0.417 | 0.478 0.450 | 0.430 0.431 | 0.460 0.418 |
| | Avg | **0.339 0.366** | 0.347 0.372 | 1.142 0.782 | 0.369 0.378 | 0.348 0.375 | 0.347 0.375 | 0.360 0.388 | 0.357 0.379 | 0.355 0.378 | 0.398 0.405 | 0.345 0.377 | 0.353 0.382 | 0.407 0.410 | 0.351 0.381 | 0.400 0.406 | 0.373 0.392 | 0.406 0.385 |
| ETTm2 | 96 | **0.157 0.244** | 0.164 0.250 | 2.041 1.073 | 0.170 0.245 | 0.164 0.254 | 0.163 0.252 | 0.176 0.265 | 0.167 0.260 | 0.161 0.251 | 0.179 0.263 | 0.168 0.261 | 0.166 0.256 | 0.187 0.267 | 0.166 0.256 | 0.187 0.267 | 0.189 0.277 | 0.205 0.273 |
| | 192 | **0.213 0.285** | 0.218 0.288 | 2.249 1.112 | 0.229 0.291 | 0.223 0.295 | 0.216 0.290 | 0.231 0.305 | 0.224 0.303 | 0.215 0.289 | 0.250 0.309 | 0.215 0.289 | 0.223 0.296 | 0.250 0.309 | 0.222 0.293 | 0.249 0.309 | 0.241 0.315 | 0.275 0.316 |
| | 336 | 0.269 **0.322** | 0.271 0.322 | 2.568 1.238 | 0.303 0.343 | 0.279 0.330 | 0.268 0.324 | 0.282 0.338 | 0.281 0.342 | **0.267** 0.326 | 0.312 0.349 | 0.278 0.337 | 0.274 0.329 | 0.311 0.348 | 0.272 0.324 | 0.321 0.351 | 0.286 0.348 | 0.329 0.350 |
| | 720 | 0.351 0.377 | 0.361 0.380 | 2.720 1.287 | 0.373 0.399 | 0.359 0.383 | 0.420 0.422 | 0.373 0.403 | 0.397 0.421 | 0.352 0.383 | 0.411 0.406 | **0.341 0.378** | 0.362 0.385 | 0.412 0.408 | 0.351 0.381 | 0.408 0.403 | 0.375 0.402 | 0.437 0.411 |
| | Avg | **0.248 0.307** | 0.254 0.310 | 2.395 1.177 | 0.269 0.320 | 0.256 0.315 | 0.267 0.322 | 0.263 0.324 | 0.267 0.332 | **0.249** 0.312 | 0.288 0.332 | 0.250 0.316 | 0.256 0.317 | 0.288 0.332 | 0.253 0.314 | 0.291 0.333 | 0.273 0.336 | 0.311 0.337 |
| Wins | | **11 16** | 0 2 | 0 0 | 0 2 | 2 2 | 0 0 | 1 0 | 0 0 | 5 1 | 0 0 | **8 3** | 1 0 | 0 0 | 2 3 | 0 0 | 0 1 | 0 0 |

[*] Zero-shot forecasting.   [†] Taken from Wu et al. [2022a].   [‡] Traffic/PEMS are often used during pre-training [Liu et al., 2025]. Thus, no zero-shot results are available.

# F   Full Results on GIFT-Eval

This section completes Tab. 2 by reporting the full GIFT-Eval results: the overall leaderboard in Tab. 8, the multivariate and univariate subsets in Tab. 9 and Tab. 10, and the domain-wise Top-10 in Tab. 11. We follow the official GIFT-Eval protocol and scoring. The tables report the aggregate CRPS and MASE, and ranks are computed by the CRPS aggregate (lower is better). Values reflect the online leaderboard at the time of submission.

Across all datasets, xLSTM-Mixer ranks 2 out of 39 models with CRPS 0.512 and MASE 0.780 (Tab. 8) and is the strongest supervised method without pretraining. On the univariate and multivariate subsets, xLSTM-Mixer also ranks 2 with CRPS 0.543 and 0.474, respectively (Tab. 9, Tab. 10). These results are consistent with the other evaluations and show that the method transfers well to probabilistic forecasting with a simple quantile head. In particular, because the multivariate CRPS is lower than the univariate CRPS (0.474 vs. 0.543), this further confirms our previous observations and design decision that mixing of variates is beneficial, suggesting that xLSTM-Mixer effectively shares information across related series (e.g., common seasonality or shocks) without domain-specific tuning in the quantile head.

The domain breakdown indicates robust behavior across heterogeneous sources. The Top-10 per-domain tables show a first place in Healthcare and typically top-five performance across the remaining domains (Tab. 11). This suggests that the model adapts well to different frequencies, scales, and noise levels without domain-specific architectural changes.

Table 8: **GIFT-Eval Overall Leaderboard (full)**. Lower is better.

| Model | MASE ↓ | CRPS ↓ | Rank (CRPS) ↓ |
|---|---|---|---|
| TiRex | 0.724 | 0.498 | 1 |
| **xLSTM-Mixer (ours)** | 0.780 | 0.510 | 2 |
| TEMPO_ensemble | 0.862 | 0.514 | 3 |
| Toto_Open_Base_1.0 | 0.750 | 0.517 | 4 |
| TabPFN-TS | 0.771 | 0.544 | 5 |
| YingLong_300m | 0.798 | 0.548 | 6 |
| timesfm_2_0_500m | 0.758 | 0.550 | 7 |
| YingLong_110m | 0.809 | 0.557 | 8 |
| sundial_base_128m | 0.750 | 0.559 | 9 |
| YingLong_50m | 0.822 | 0.567 | 10 |
| chronos_bolt_base | 0.808 | 0.574 | 11 |
| chronos_bolt_small | 0.822 | 0.577 | 12 |
| TTM-R2-Finetuned | 0.756 | 0.583 | 13 |
| PatchTST | 0.849 | 0.587 | 14 |
| Moirai_large | 0.875 | 0.599 | 15 |
| TFT | 0.915 | 0.605 | 16 |
| YingLong_6m | 0.880 | 0.609 | 17 |
| Moirai_base | 0.901 | 0.610 | 18 |
| iTransformer | 0.893 | 0.620 | 19 |
| Chronos_large | 0.870 | 0.647 | 20 |
| Moirai_small | 0.946 | 0.650 | 21 |
| Chronos_base | 0.876 | 0.652 | 22 |
| Chronos_small | 0.892 | 0.663 | 23 |
| TimesFM | 1.077 | 0.680 | 24 |
| VisionTS | 0.863 | 0.755 | 25 |
| TIDE | 1.091 | 0.772 | 26 |
| N-BEATS | 0.938 | 0.816 | 27 |
| DLinear | 1.061 | 0.846 | 28 |
| DeepAR | 1.343 | 0.853 | 29 |
| TTM-R2-Zeroshot | 1.020 | 0.873 | 30 |
| Lag-Llama | 1.228 | 0.880 | 31 |
| TTM-R1-Zeroshot | 1.079 | 0.891 | 32 |
| Auto_Arima | 1.074 | 0.912 | 33 |
| Timer | 1.136 | 0.970 | 34 |
| seasonal_naive | 1.000 | 1.000 | 35 |
| Auto_Theta | 1.090 | 1.244 | 36 |
| Naive | 1.270 | 1.591 | 37 |
| Crossformer | 2.574 | 1.637 | 38 |
| Auto_ETS | 1.212 | 7.489 | 39 |

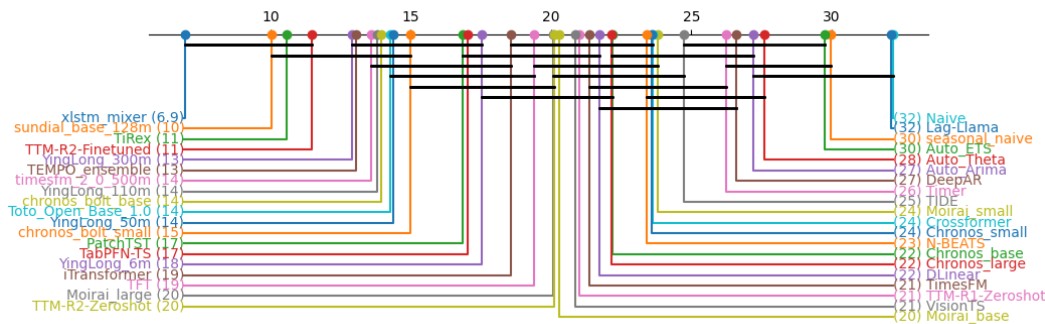

Figure 5: **xLSTM-Mixer is statistically significantly better than most baselines.** Shown is the critical difference diagram for the MSE. Horizontal connected bars indicate that those methods are not significantly different at $p = 0.05$.

Table 9: **GIFT-Eval Multivariate Leaderboard (full).** Lower is better.

| Model | MASE ↓ | CRPS ↓ | Rank (CRPS) ↓ |
|---|---|---|---|
| TEMPO_ensemble | 0.752 | 0.427 | 1 |
| **xLSTM-Mixer (ours)** | 0.839 | 0.473 | 2 |
| Toto_Open_Base_1.0 | 0.767 | 0.484 | 3 |
| TiRex | 0.771 | 0.490 | 4 |
| YingLong_300m | 0.840 | 0.519 | 5 |
| YingLong_110m | 0.830 | 0.520 | 6 |
| YingLong_50m | 0.853 | 0.530 | 7 |
| sundial_base_128m | 0.778 | 0.530 | 8 |
| TabPFN-TS | 0.818 | 0.544 | 9 |
| timesfm_2_0_500m | 0.803 | 0.553 | 10 |
| PatchTST | 0.906 | 0.556 | 11 |
| YingLong_6m | 0.905 | 0.566 | 12 |
| iTransformer | 0.940 | 0.589 | 13 |
| TTM-R2-Finetuned | 0.808 | 0.591 | 14 |
| chronos_bolt_small | 0.939 | 0.600 | 15 |
| chronos_bolt_base | 0.925 | 0.604 | 16 |
| TFT | 1.071 | 0.610 | 17 |
| Moirai_large | 1.023 | 0.635 | 18 |
| Moirai_base | 1.054 | 0.636 | 19 |
| Moirai_small | 1.021 | 0.640 | 20 |
| Chronos_large | 1.005 | 0.680 | 21 |
| Chronos_small | 1.026 | 0.684 | 22 |
| Chronos_base | 1.013 | 0.685 | 23 |
| TimesFM | 1.497 | 0.717 | 24 |
| VisionTS | 0.887 | 0.721 | 25 |
| N-BEATS | 0.998 | 0.790 | 26 |
| Lag-Llama | 1.221 | 0.798 | 27 |
| TIDE | 1.284 | 0.812 | 28 |
| DLinear | 1.228 | 0.817 | 29 |
| Crossformer | 1.479 | 0.825 | 30 |
| TTM-R2-Zeroshot | 1.072 | 0.833 | 31 |
| TTM-R1-Zeroshot | 1.186 | 0.856 | 32 |
| Timer | 1.141 | 0.883 | 33 |
| DeepAR | 1.908 | 0.989 | 34 |
| seasonal_naive | 1.000 | 1.000 | 35 |
| Auto_Arima | 1.318 | 1.032 | 36 |
| Auto_Theta | 1.022 | 1.141 | 37 |
| Naive | 1.167 | 1.455 | 38 |
| Auto_ETS | 1.346 | 4.996 | 39 |

Table 10: **GIFT-Eval Univariate Leaderboard (full).** Lower is better.

| Model | MASE ↓ | CRPS ↓ | Rank (CRPS) ↓ |
|---|---|---|---|
| TiRex | 0.688 | 0.505 | 1 |
| **xLSTM-Mixer (ours)** | 0.737 | 0.541 | 2 |
| TabPFN-TS | 0.735 | 0.544 | 3 |
| Toto_Open_Base_1.0 | 0.737 | 0.545 | 4 |
| timesfm_2_0_500m | 0.724 | 0.549 | 5 |
| chronos_bolt_base | 0.725 | 0.552 | 6 |
| chronos_bolt_small | 0.739 | 0.559 | 7 |
| Moirai_large | 0.773 | 0.572 | 8 |
| YingLong_300m | 0.766 | 0.573 | 9 |
| TTM-R2-Finetuned | 0.717 | 0.576 | 10 |
| sundial_base_128m | 0.729 | 0.583 | 11 |
| Moirai_base | 0.795 | 0.589 | 12 |
| YingLong_110m | 0.793 | 0.589 | 13 |
| TEMPO_ensemble | 0.960 | 0.596 | 14 |
| YingLong_50m | 0.799 | 0.598 | 15 |
| TFT | 0.808 | 0.601 | 16 |
| PatchTST | 0.805 | 0.613 | 17 |
| Chronos_large | 0.775 | 0.622 | 18 |
| Chronos_base | 0.780 | 0.627 | 19 |
| YingLong_6m | 0.861 | 0.646 | 20 |
| Chronos_small | 0.797 | 0.647 | 21 |
| iTransformer | 0.857 | 0.647 | 22 |
| TimesFM | 0.829 | 0.652 | 23 |
| Moirai_small | 0.890 | 0.659 | 24 |
| TIDE | 0.959 | 0.741 | 25 |
| DeepAR | 1.016 | 0.758 | 26 |
| VisionTS | 0.845 | 0.783 | 27 |
| Auto_Arima | 0.912 | 0.826 | 28 |
| N-BEATS | 0.892 | 0.837 | 29 |
| DLinear | 0.944 | 0.869 | 30 |
| TTM-R2-Zeroshot | 0.980 | 0.907 | 31 |
| TTM-R1-Zeroshot | 1.001 | 0.920 | 32 |
| Lag-Llama | 1.233 | 0.952 | 33 |
| seasonal_naive | 1.000 | 1.000 | 34 |
| Timer | 1.131 | 1.047 | 35 |
| Auto_Theta | 1.147 | 1.332 | 36 |
| Naive | 1.358 | 1.709 | 37 |
| Crossformer | 4.000 | 2.824 | 38 |
| Auto_ETS | 1.115 | 10.337 | 39 |

Table 11: **GIFT-Eval Domain Leaderboards (Top-10 per domain)**. Lower is better.

| Domain | Model | MASE ↓ | CRPS ↓ | Rank ↓ |
|---|---|---|---|---|
| *Econ/Fin* | timesfm_2_0_500m | 0.640 | 0.580 | 1 |
| | TiRex | 0.746 | 0.709 | 2 |
| | chronos_bolt_small | 0.816 | 0.743 | 3 |
| | TimesFM | 0.824 | 0.761 | 4 |
| | chronos_bolt_base | 0.799 | 0.762 | 5 |
| | Moirai_large | 0.845 | 0.778 | 6 |
| | TabPFN-TS | 0.810 | 0.785 | 7 |
| | Chronos_base | 0.783 | 0.798 | 8 |
| | **xLSTM-Mixer (ours)** | 0.975 | 0.805 | 9 |
| | Chronos_large | 0.782 | 0.806 | 10 |
| *Energy* | TiRex | 0.820 | 0.589 | 1 |
| | TEMPO_ensemble | 1.063 | 0.613 | 2 |
| | YingLong_300m | 0.870 | 0.627 | 3 |
| | Toto_Open_Base_1.0 | 0.876 | 0.628 | 4 |
| | **xLSTM-Mixer (ours)** | 0.881 | 0.633 | 5 |
| | chronos_bolt_base | 0.846 | 0.640 | 6 |
| | TabPFN-TS | 0.879 | 0.641 | 7 |
| | sundial_base_128m | 0.839 | 0.645 | 8 |
| | YingLong_110m | 0.907 | 0.651 | 9 |
| | chronos_bolt_small | 0.864 | 0.656 | 10 |
| *Healthcare* | **xLSTM-Mixer (ours)** | 0.522 | 0.403 | 1 |
| | TabPFN-TS | 0.576 | 0.450 | 2 |
| | TTM-R2-Finetuned | 0.559 | 0.460 | 3 |
| | Toto_Open_Base_1.0 | 0.625 | 0.467 | 4 |
| | Chronos_large | 0.599 | 0.472 | 5 |
| | TiRex | 0.628 | 0.473 | 6 |
| | timesfm_2_0_500m | 0.597 | 0.481 | 7 |
| | Chronos_base | 0.644 | 0.513 | 8 |
| | Chronos_small | 0.607 | 0.525 | 9 |
| | chronos_bolt_small | 0.671 | 0.541 | 10 |
| *Nature* | TEMPO_ensemble | 0.601 | 0.317 | 1 |
| | chronos_bolt_base | 0.667 | 0.327 | 2 |
| | TiRex | 0.686 | 0.328 | 3 |
| | Toto_Open_Base_1.0 | 0.736 | 0.348 | 4 |
| | timesfm_2_0_500m | 0.624 | 0.350 | 5 |
| | chronos_bolt_small | 0.704 | 0.351 | 6 |
| | YingLong_300m | 0.746 | 0.354 | 7 |
| | YingLong_110m | 0.743 | 0.356 | 8 |
| | sundial_base_128m | 0.703 | 0.361 | 9 |
| | YingLong_50m | 0.748 | 0.365 | 10 |

| Domain | Model | MASE ↓ | CRPS ↓ | Rank ↓ |
|---|---|---|---|---|
| *Sales* | TiRex | 0.682 | 0.415 | 1 |
| | TabPFN-TS | 0.695 | 0.419 | 2 |
| | timesfm_2_0_500m | 0.700 | 0.419 | 2 |
| | TimesFM | 0.701 | 0.421 | 4 |
| | chronos_bolt_base | 0.694 | 0.422 | 5 |
| | Moirai_base | 0.695 | 0.424 | 6 |
| | chronos_bolt_small | 0.696 | 0.424 | 6 |
| | Toto_Open_Base_1.0 | 0.705 | 0.424 | 6 |
| | PatchTST | 0.691 | 0.426 | 9 |
| | iTransformer | 0.699 | 0.430 | 10 |
| *Transport* | Moirai_large | 0.601 | 0.451 | 1 |
| | TiRex | 0.624 | 0.468 | 2 |
| | Toto_Open_Base_1.0 | 0.632 | 0.477 | 3 |
| | Moirai_base | 0.637 | 0.478 | 4 |
| | **xLSTM-Mixer (ours)** | 0.635 | 0.487 | 5 |
| | TTM-R2-Finetuned | 0.627 | 0.496 | 6 |
| | timesfm_2_0_500m | 0.645 | 0.501 | 7 |
| | sundial_base_128m | 0.634 | 0.504 | 8 |
| | YingLong_300m | 0.666 | 0.504 | 8 |
| | TFT | 0.679 | 0.514 | 10 |
| *Web/CloudOps* | TEMPO_ensemble | 0.585 | 0.387 | 1 |
| | **xLSTM-Mixer (ours)** | 0.779 | 0.457 | 2 |
| | Toto_Open_Base_1.0 | 0.694 | 0.500 | 3 |
| | TiRex | 0.716 | 0.518 | 4 |
| | sundial_base_128m | 0.695 | 0.552 | 5 |
| | PatchTST | 0.780 | 0.553 | 6 |
| | TabPFN-TS | 0.727 | 0.573 | 7 |
| | YingLong_110m | 0.814 | 0.575 | 8 |
| | iTransformer | 0.823 | 0.575 | 8 |
| | YingLong_300m | 0.846 | 0.584 | 10 |

## G Visualizing Initial Token Embeddings

Fig. 6 shows how the learned initial tokens $\eta$ reflect common patterns found in the datasets. A row-by-row inspection of the figure reveals that similar patterns are learned per dataset, albeit at different scales. Specifically, these patterns repeat in proportion to the forecast horizon. The fact that essentially the same patterns are learned for each dataset across different initializations and horizons supports the conclusion that they are data-driven and meaningful to the domain.

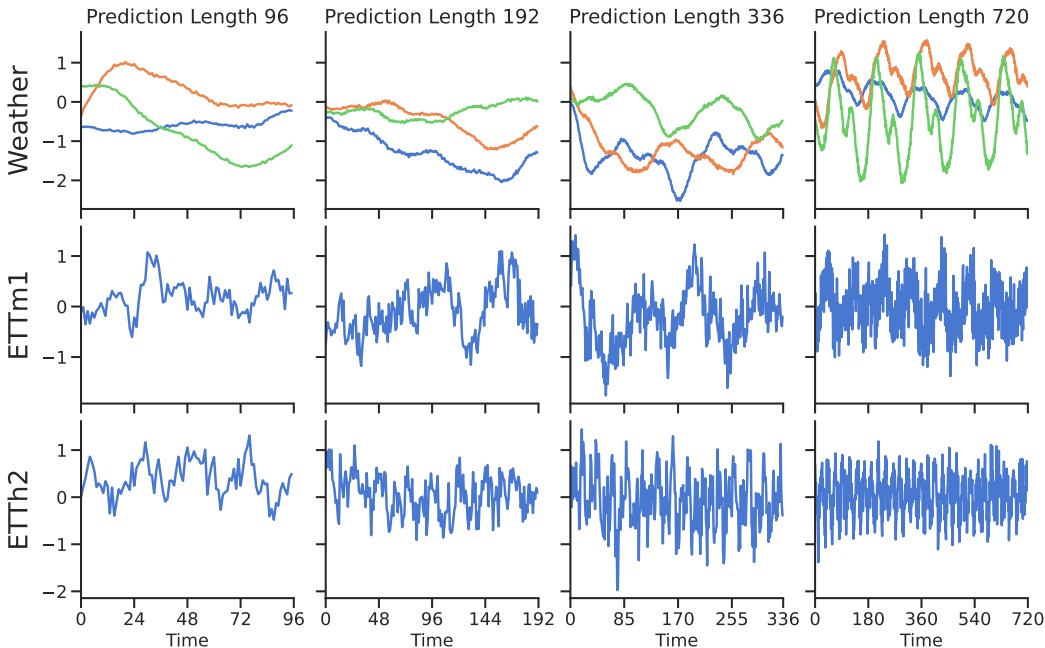

Figure 6: **Initial tokens capture dataset characteristics.** The plot illustrates the learned tokens across multiple datasets and prediction lengths. The lookback length is set to 96 for all evaluations. For clarity and the high noise levels of the data, only a single seed is shown for ETTm1 and ETTh2.

## H Learning Cross-Variate Patterns

As the sLSTM refinement blocks $\mathcal{S}$ process the variates recurrently, it is insightful to assess the extent to which inter-variate relationships are effectively captured. To this end, we adopt a perturbation-based approach to compute attributions, approximating Shapley Values through sampling. Hereby, we use a zero baseline and follow the horizon aggregation method proposed by Kraus et al. [2025], where the forecasts over the entire horizon are aggregated into a single scalar, which serves as the target for the attribution computation. We visualize these Shapley-based feature attribution scores, illustrating the degree to which each output variate of the xLSTM-Mixer depends on each input variate. Fig. 7 demonstrates the ability of xLSTM-Mixer to model cross-variate relationships effectively. Due to the design of the sLSTM refinement module, which strides over the variates, each variate can only be influenced by the ones preceding it. This restriction is reflected in the attribution scores, which appear exclusively in the lower-left triangle.

xLSTM-Mixer fixes one variate ordering to learn multivariate relationships efficiently. We investigate its impact by randomly permuting variate orders and comparing results with the baseline. Tab. 12 shows the results for four such permutations over four horizons on the Weather and ETTm1 datasets. We observe that the specific ordering does play some role in forecasting performance. However, the standard ordering provided by the dataset sources already permits highly effective forecasting.

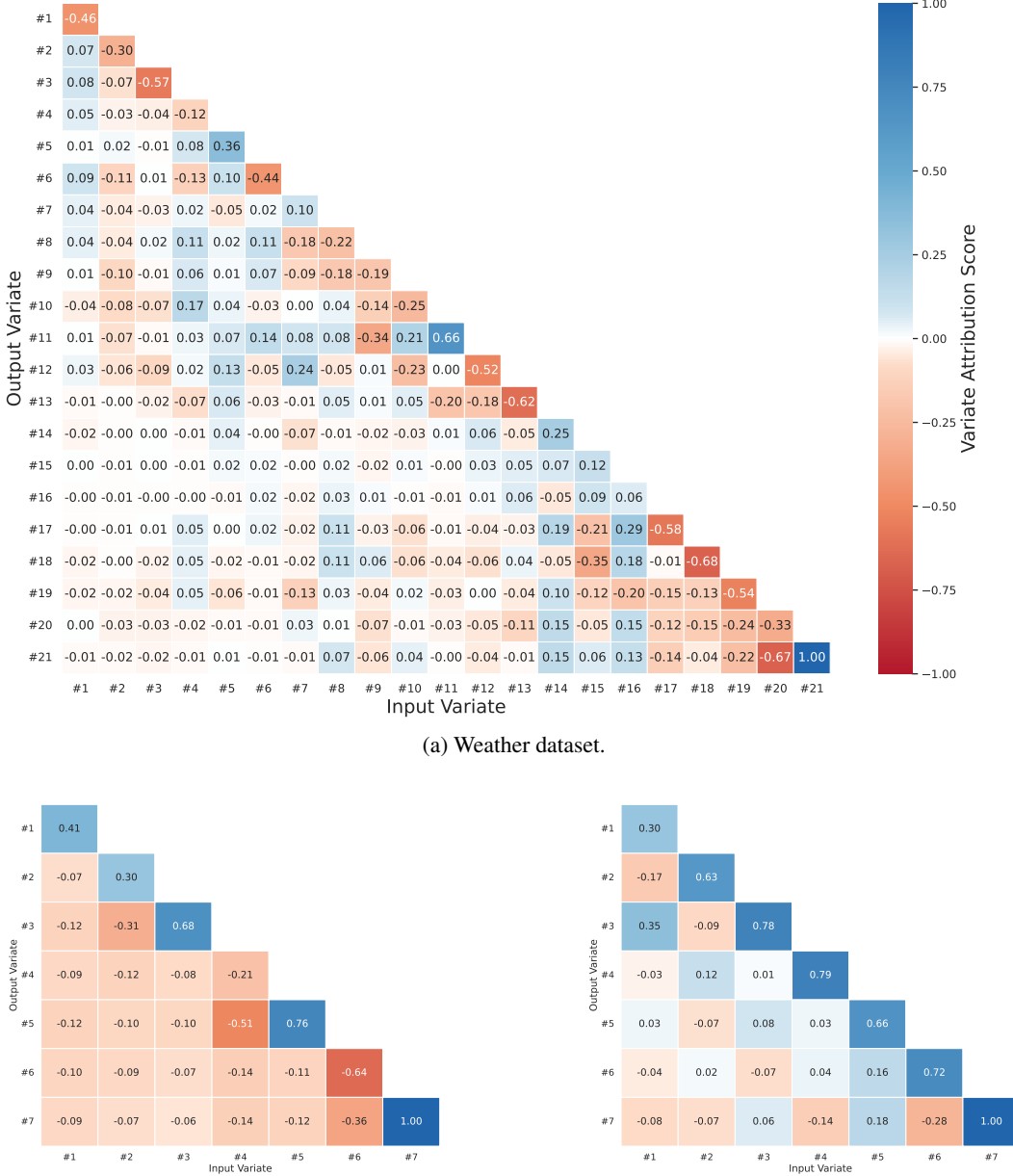

(a) Weather dataset.

(b) ETTh2 dataset.

(c) ETTm1 dataset.

Figure 7: **xLSTM-Mixer effectively learns cross-variate patterns,** as this feature attribution of each output to input variate on the Weather dataset demonstrates.

Table 12: **xLSTM-Mixer provides strong performance regardless of the variate permutations.** All measurements are averaged over three initializations each.

| | | From Dataset | | Perm. #1 | | Perm. #2 | | Perm. #3 | | Perm. #4 | |
|---|---|---|---|---|---|---|---|---|---|---|---|
| Metric | | MSE | MAE | MSE | MAE | MSE | MAE | MSE | MAE | MSE | MAE |
| Weather | 96 | 0.143 | 0.184 | 0.149 | 0.189 | 0.146 | 0.187 | 0.147 | 0.188 | 0.148 | 0.188 |
| | 192 | 0.186 | 0.226 | 0.192 | 0.229 | 0.191 | 0.229 | 0.192 | 0.229 | 0.192 | 0.230 |
| | 336 | 0.236 | 0.266 | 0.242 | 0.269 | 0.241 | 0.269 | 0.242 | 0.269 | 0.240 | 0.269 |
| | 720 | 0.310 | 0.323 | 0.310 | 0.323 | 0.310 | 0.323 | 0.310 | 0.323 | 0.310 | 0.323 |
| ETTm1 | 96 | 0.275 | 0.328 | 0.278 | 0.331 | 0.276 | 0.330 | 0.277 | 0.330 | 0.275 | 0.329 |
| | 192 | 0.319 | 0.354 | 0.321 | 0.356 | 0.321 | 0.356 | 0.320 | 0.355 | 0.319 | 0.355 |
| | 336 | 0.353 | 0.374 | 0.355 | 0.376 | 0.355 | 0.376 | 0.354 | 0.376 | 0.354 | 0.376 |
| | 720 | 0.409 | 0.407 | 0.412 | 0.409 | 0.413 | 0.410 | 0.413 | 0.410 | 0.413 | 0.410 |
| Electricity | 96 | 0.126 | 0.218 | 0.127 | 0.220 | 0.126 | 0.218 | 0.127 | 0.219 | 0.125 | 0.218 |
| | 192 | 0.144 | 0.235 | 0.145 | 0.237 | 0.144 | 0.235 | 0.145 | 0.236 | 0.144 | 0.235 |
| | 336 | 0.157 | 0.250 | 0.160 | 0.252 | 0.159 | 0.251 | 0.157 | 0.248 | 0.159 | 0.250 |
| | 720 | 0.183 | 0.276 | 0.230 | 0.315 | 0.225 | 0.312 | 0.206 | 0.295 | 0.218 | 0.306 |

# I  Ensembling over More than Two Views

Tab. 13 shows the results for ensembling varying numbers of views. Note that even if $E > 2$, the first two views are always fixed to be forward and backward.

As we can see, the base version of using two views is the overall best choice. We observed this on multiple datasets and show ETTh2 as a representative sample.

Table 13: **Ensembling over more than two views does not yield further benefits.** Results are for different numbers of views $E$ for varying forecast horizons $H$ on ETTh2. The values in parentheses represent the relative change over $E = 1$ (only forward view), where lower numbers are better. $E = 2$ is the base variant of xLSTM-Mixer, showing the strongest improvement each.

| $E$ | $H = 96$ MSE | MAE | $H = 192$ MSE | MAE | $H = 336$ MSE | MAE | $H = 720$ MSE | MAE |
|---|---|---|---|---|---|---|---|---|
| 1 | 0.274 | 0.330 | 0.345 | 0.375 | 0.377 | 0.402 | 0.397 | 0.426 |
| 2 | 0.267 -2.55% | 0.329 -0.30% | 0.338 -2.03% | 0.375 +0.00% | 0.367 -2.65% | 0.401 -0.25% | 0.388 -2.27% | 0.424 -0.47% |
| 5 | 0.273 -0.36% | 0.333 +0.91% | 0.340 -1.45% | 0.380 +1.33% | 0.374 -0.80% | 0.408 +1.49% | 0.393 -1.01% | 0.430 +0.94% |
| 8 | 0.275 +0.36% | 0.336 +1.82% | 0.344 -0.29% | 0.382 +1.87% | 0.373 -1.06% | 0.407 +1.24% | 0.408 +2.77% | 0.439 +3.05% |
| 10 | 0.274 +0.00% | 0.336 +1.82% | 0.345 +0.00% | 0.383 +2.13% | 0.372 -1.33% | 0.405 +0.75% | 0.411 +3.53% | 0.441 +3.52% |

# J Outlook: Classification

Tab. 14 compares xLSTM-Mixer to common time series classification models and standard benchmark datasets. Summarizing the individual results, xLSTM-Mixer is on par with the best model, ModernTCN [Donghao and Xue, 2023].

To adapt xLSTM-Mixer for classification, we replace the final regression projection layer with a single fully connected classification head followed by a softmax activation. We train this variant end-to-end on labeled sequences using cross-entropy loss, applying the same data preprocessing and augmentation pipeline as in our forecasting experiments.

These results highlight xLSTM-Mixer's flexibility: with minimal architectural changes and no specialized classification tricks, it achieves state-of-the-art performance while preserving the same core building blocks used for forecasting. This unified design suggests that xLSTM-Mixer can serve as a general backbone for a wide range of sequence modeling tasks.

Table 14: **xLSTM-Mixer is effective at time series classification.** We report the averaged accuracy in percent. Adapted from Wu et al. [2022a].

| | Classical | | | | Recurrent | | SSM | Transformer | | | | | | | | | | MLP | | | Convolutional | |
|---|---|---|---|---|---|---|---|---|---|---|---|---|---|---|---|---|---|---|---|---|---|---|
| Datasets | DTW [1994] | XGBoost [2016] | Rocket [2020] | xLSTM-Mixer | LSTM [1997] | LSTNet [2018] | S4 [2022] | Trans. [2017] | Re. [2020] | In. [2021] | Pyra. [2021] | Auto. [2021] | Station. [2022] | FED. [2022] | ETS. [2022] | Flow. [2022b] | iTransf. [2023] | DLin. [2023] | LightTS [2023] | TiDE [2023] | TimesNet [2022a] | M.TCN [2023] |
| EthanolConcentration | 32.3 | 43.7 | 45.2 | 31.7 | 32.3 | 39.9 | 31.1 | 32.7 | 31.9 | 31.6 | 30.8 | 31.6 | 32.7 | 28.1 | 31.2 | 33.8 | 28.1 | 32.6 | 29.7 | 27.1 | 35.7 | 36.3 |
| FaceDetection | 52.9 | 63.3 | 64.7 | 68.9 | 57.7 | 65.7 | 66.7 | 67.3 | 68.6 | 67.0 | 65.7 | 68.4 | 68.0 | 66.0 | 66.3 | 67.6 | 66.3 | 68.0 | 67.5 | 65.3 | 68.6 | 70.8 |
| Handwriting | 28.6 | 15.8 | 58.8 | 31.8 | 15.2 | 25.8 | 24.6 | 32.0 | 27.4 | 32.8 | 29.4 | 36.7 | 31.6 | 28.0 | 32.5 | 33.8 | 24.2 | 27.0 | 26.1 | 23.2 | 32.1 | 30.6 |
| Heartbeat | 71.7 | 73.2 | 75.6 | 77.7 | 72.2 | 77.1 | 72.7 | 76.1 | 77.1 | 80.5 | 75.6 | 74.6 | 73.7 | 73.7 | 71.2 | 77.6 | 75.6 | 75.1 | 75.1 | 74.6 | 78.0 | 77.2 |
| JapaneseVowels | 94.9 | 86.5 | 96.2 | 97.5 | 79.7 | 98.1 | 98.4 | 98.7 | 97.8 | 98.9 | 98.4 | 96.2 | 99.2 | 98.4 | 95.9 | 98.9 | 96.6 | 96.2 | 96.2 | 95.6 | 98.4 | 98.8 |
| PEMS-SF | 71.1 | 98.3 | 75.1 | 91.5 | 39.9 | 86.7 | 86.1 | 82.1 | 82.7 | 81.5 | 83.2 | 82.7 | 87.3 | 80.9 | 86.0 | 83.8 | 87.9 | 75.1 | 88.4 | 86.9 | 89.6 | 89.1 |
| SelfRegulationSCP1 | 77.7 | 84.6 | 90.8 | 93.6 | 68.9 | 84.0 | 90.8 | 92.2 | 90.4 | 90.1 | 88.1 | 84.0 | 89.4 | 88.7 | 89.6 | 92.5 | 90.2 | 87.3 | 89.8 | 89.2 | 91.8 | 93.4 |
| SelfRegulationSCP2 | 53.9 | 48.9 | 53.3 | 59.8 | 46.6 | 52.8 | 52.2 | 53.9 | 56.7 | 53.3 | 53.3 | 50.6 | 57.2 | 54.4 | 55.0 | 56.1 | 54.4 | 50.5 | 51.1 | 53.4 | 57.2 | 60.3 |
| SpokenArabicDigits | 96.3 | 69.6 | 71.2 | 99.3 | 31.9 | 100.0 | 100.0 | 98.4 | 97.0 | 100.0 | 99.6 | 100.0 | 100.0 | 100.0 | 100.0 | 98.8 | 96.0 | 81.4 | 100.0 | 95.0 | 99.0 | 98.7 |
| UWaveGestureLibrary | 90.3 | 75.9 | 94.4 | 89.6 | 41.2 | 87.8 | 85.9 | 85.6 | 85.6 | 85.6 | 83.4 | 85.9 | 87.5 | 85.3 | 85.0 | 86.6 | 85.9 | 82.1 | 80.3 | 84.9 | 85.3 | 86.7 |
| Average Accuracy | 67.0 | 66.0 | 72.5 | 74.1 | 48.6 | 71.8 | 70.9 | 71.9 | 71.5 | 72.1 | 70.8 | 71.1 | 72.7 | 70.7 | 71.0 | 73.0 | 70.5 | 67.5 | 70.4 | 69.5 | 73.6 | **74.2** |

