# OpenReview forum: "xLSTM-Mixer: Multivariate Time Series Forecasting by Mixing via Scalar Memories"
_NeurIPS.cc/2025/Conference — NeurIPS 2025 poster_

### Official Review · Reviewer_cGWV · 2025-06-19

**Clarity:** 2
**Significance:** 3
**Originality:** 3
**Rating:** 5
**Confidence:** 3

**Summary:**

As fall we know, the several patterns both within and between temporal and multivariate components is crucial for reliable predictions. In this work, the authors introduce xLSTM-Mixer method, a model designed to effectively integrate temporal sequences, joint time-variate information, and multiple perspectives for robust forecasting. The proposed approach begins with a linear forecast shared across variates, which is then refined by xLSTM blocks. They serve as key elements for modeling the complex dynamics of challenging time series data. xLSTM-Mixer ultimately reconciles two distinct views to produce the final forecast. This is a model design scheme from an interesting perspective.

**Questions:**

Please see Weaknesses!

**Ethical Concerns:**

["NO or VERY MINOR ethics concerns only"]

**Final Justification:**

The author has addressed all my concerns, mainly by elaborating on the issue of interpretability.

**Limitations:**

Please see Weaknesses!

**Quality:**

3

**Strengths And Weaknesses:**

Strengths:
1. It seems to be the first attempt to integrate recurrent models (xLSTM) with mixing architectures (time/variate/view mixing). The three-stage design (linear forecast → sLSTM refinement → view reconciliation) is novel and well-justified.
2. Evaluated on 7 time series forecasting benchmark datasets (Weather, Electricity, ETT, etc.) across 4 horizons (96–720 steps). Outperforms 20+ SOTA baselines (Table 1), achieving best results in 11/28 MSE and 16/28 MAE cases.
3. Uses exponential gating and memory mixing (i.e., Eqs. 1–8) to capture cross-variate dependencies in the xLSTM network. Multi-head recurrence matrices enhance efficiency without sacrificing expressiveness.

Weaknesses:
1. Ablation (in Table 2) shows omitting the initial token embedding still yields competitive results. Its necessity and interpretability (Fig. 5) require deeper analysis.

---

> ### Author Rebuttal · Authors · 2025-07-31
>
> Dear Reviewer cGWV,
>
> Thank you for your strongly affirmative review and for recognizing the novelty, clarity, and empirical efficacy of xLSTM-Mixer.
>
> We are happy to provide further insights into the initial token embedding:
>
> **W1 (necessity):** The benefit of learning $\mathbf\eta$ can be observed by comparing ablation #7 to the base xLSTM-Mixer model #1 in Table 2. Model #1 consistently performs at least as well as #7 and, in 13 out of 16 cases, learning the initial token embedding yields strictly better results. While the effect is not dramatic, it offers a consistent benefit: on average, learning the initial token embedding reduces MAE by 0.4% and MSE by 0.7% across all eight settings.
>
> **W1 (interpretability):** We are pleased to expand on the interpretation of Fig. 5, as outlined in lines 237–240. A row-by-row inspection of the figure reveals that similar patterns are learned per dataset, albeit at different scales. Specifically, these patterns repeat in proportion to the forecast horizon. The fact that essentially the same patterns are learned for the same datasets across different initializations and horizons supports the conclusion that they are data-driven and meaningful to the domain.
>
> We also invite you to review our response to Reviewer 67JL, where we significantly expanded our empirical analysis on the GIFT-eval benchmark, reaching impressive leaderboard placements (Q2), conducted statistical significance tests (Q4), and extended the experiments on variate orderings (Q5).
>
> Thank you again for your overwhelmingly positive review. In light of this, we would greatly appreciate it if you could consider raising your score to reflect your assessment.
>
> Best
>
> The Authors

---

> > ### Comment · Reviewer_cGWV · 2025-08-04
> > **Response regarding rebuttal**
> >
> > Dear Authors,
> >
> > Thank you for your further reply. I'm glad to see that the concern has been resolved. I have decided to increase the score to 5.

---

> ### Author Response · Authors · 2025-08-04
>
> Dear Reviewer cGWV,
>
> Thank you for your helpful review and positive assessment of the work. Your comments sharpened our analysis of the initial embedding and strengthened the paper. The camera-ready version will reflect these improvements.
>
> Bests
>
> The Authors

---

### Official Review · Reviewer_fZrP · 2025-07-02

**Clarity:** 3
**Significance:** 2
**Originality:** 2
**Rating:** 4
**Confidence:** 3

**Summary:**

The authors introduces the xLSTM-Mixer, a new model that leverages scalar LSTM units in a multi-stage setup to blend temporal and variate information for more accurate and efficient multivariate time series forecasting. The architecture moves from a channel-independent linear forecast, through a joint mixing phase with sLSTM blocks, to a final stage that reconciles forecasts from both original and reversed data views, which really boosts generalization and stability. Benchmarks show xLSTM-Mixer consistently outperforms top models in accuracy and efficiency, especially for long-term forecasting, and it’s robust across short-term, classification, and probabilistic tasks—making it a strong, practical advance for time series modeling.

**Questions:**

1. The use of forward and reversed embeddings with shared weights for forecast reconciliation is interesting. Can the authors more clearly explain how this approach differs fundamentally from data augmentation or bidirectional recurrence? What is the theoretical or empirical basis for expecting this mixing to yield better generalization?

Suggestion: A more detailed ablation isolating this component—including results without reversal or with separate weights—would help clarify whether the gains come from view diversity, weight sharing, or both.

2.  The experiments are conducted on standard benchmark datasets. Can the authors provide results or at least discussion on how xLSTM-Mixer might generalize to noisier or domain-specific settings (e.g., irregular sampling, missing data, non-stationary series)?

Suggestion: Including at least one example or case study from a real-world applied domain (e.g., energy load forecasting, medical monitoring) would help demonstrate the model’s robustness and practical utility.


3. Given the layered and multistage nature of the architecture, how feasible is it to interpret model outputs in operational settings? Are there mechanisms in xLSTM-Mixer that support interpretability?

Suggestion: It would be good to have a section discussing the trade-offs between predictive accuracy and interpretability, and any practical tools available for model explainability.

4. The authors suggest all components contribute positively, but their quantitative contribution is not deeply discussed. Could the authors provide more granular comparisons—e.g., percentage drop in MAE/MSE for each omitted feature?

Suggestion: It would be good to include a clearer, perhaps tabular, breakdown of performance degradation when omitting each core block (e.g., RevIN, view mixing, soft prompt, variate striding).

5. The author chooses to exclude mLSTM blocks in favor of sLSTM due to their “independent treatment of sequence elements.” Could the authors expand on why sLSTM is uniquely suited for joint time-variate mixing?

Suggestion: a short side-by-side comparison (theoretical or empirical) would  be great.

**Ethical Concerns:**

["Major Concern: Environmental impact"]

**Limitations:**

The authors do not explicitly address limitations of xLSTM-Mixer in the current version of the paper. While the model is clearly presented as efficient and performant, some critical factors are missing:

1. Limitations in Real-World Applicability: Please provide how xLSTM-Mixer performs in real-world scenarios involving irregular sampling, missing data, or non-stationary trends, which are common in many industrial time series applications.

2. Resource and Deployment Context: The author highlights memory and computational efficiency, but should also reflect on how this might enable broader deployment, including in settings where technical oversight or monitoring is limited (e.g., automated decision-making in low-resource regions), and what risks that might entail.

3. Societal Impact Consideration: Given that time series forecasting models can be deployed in sensitive domains (e.g., energy systems, financial markets, healthcare), the authors should consider discussing how model biases, mispredictions, or over-reliance on automation might affect stakeholders.

**Quality:**

3

**Strengths And Weaknesses:**

This  paper stands out for its thorough empirical validation and thoughtful design, showcasing impressive forecasting accuracy and efficiency across diverse datasets and tasks, thanks to best practices like RevIN normalization and a clever three-stage architecture that mixes both original and reversed data views. While the model’s efficiency and systematic approach are big strengths—especially compared to Transformer-based methods—the complexity and dense terminology may make it less accessible to those outside deep learning, and key details being in the appendices doesn’t help. The novelty lies more in the integration of existing ideas than in brand-new algorithms, and the focus on standard benchmarks leaves some questions about real-world performance. Still, it’s a strong, timely contribution, and with more emphasis on clarity and practical deployment, its impact could be even bigger

---

> ### Author Rebuttal · Authors · 2025-07-31
>
> Dear Reviewer fZrP,
>
> We thank the reviewer for recognizing the thoughtful design and thorough empirical validation of xLSTM-Mixer.
>
> **W1:** Thank you for mentioning this interesting perspective.
> This multi-view prediction is primarily related to multi-task learning (Zhang et al., 2022), which is similar to data augmentation in its goal to regularize training. Mixing operations can be interpreted as a latent data augmentation or alternatively as ensembling with weight-sharing as suggested by Reviewer 67JL. We can imagine these perspectives as a guide for future improvements to xLSTM-Mixer, and we will add them to Sec. 3.3.
>
> **W2:** Yes, we indeed conducted the very extensive real-world benchmark GIFT-Eval (Aksu et al., 2024) encompassing 97 scenarios. It also includes missing values as you suggested, empirically strengthening our contribution. Furthermore, multiple benchmark datasets are also non-stationary. We also note that the existing benchmark datasets already exhibit a diverse range of substantial levels of noise. This is also shown by the Hurst exponent in Appendix A, Tab. 3.
> We added this discussion to the limitations section of the manuscript as laid out at the end of this response.
>
> **W3:** Explainable AI is indeed an important aspect, and often at conflict with model fidelity.
> On one hand, the multistage approach makes certain interpretations harder, like for any deep learning model. See also the limitations section at the end of this response.
> On the other hand, it permits, for instance, interpreting the initial token embeddings as shown in ll. 234ff and in Appendix D.
> Overall, attribution methods, such as Integrated Gradients and SHAP, still function as normal, permitting inspection of the learned patterns, such as shown in ll. 266ff and in Appendix E.
>
> **W4:** Highlighting the differences of the variants in the ablation by including performance drops relative to the full xLSTM-Mixer baseline (#1) is a very useful suggestion. We will improve Sec. 4.4 by adding the following information.
> > Ablating components of xLSTM-Mixer each causes both error metrics to increase, entailing that they contribute positively to the overall architecture.
> > Specifically, ommiting xLSTM (only having LSTM, #3) raises the MAE by 6.2% / the MSE by 7.0%, variate recurrence (#5) by 4.3%/4.7%, learning initial token embedding (#7) by 0.4%/0.7%, view mixing (#8) by 0.6%/0.7%, and time mixing (#11) by 2.7%/3.1%.
>
> **W5:** Thank you for mentioning this. Reviewer V19p raised a similar point. Please refer to the response of Reviewr V19p **Q1**, where we clarify and give an intuition why state‑tracking via memory mixing in sLSTM is indispensable, whereas mLSTM's design cannot provide this functionality.
>
> **Comments on Limitations:**
> We welcome your comments on the limitations of xLSTM-Mixer. We added the following to the paper: (1) A dedicated limitations paragraph in Section 6 and (2) a dedicated section on broader societal impact as encouraged by NeurIPS. We attached them both to this response:
>
> > **Limitations.**
> > While xLSTM‑Mixer achieves strong accuracy with limited compute, certain assumptions need to be met for it to be applicable. Specifically, it assumes that all variables are sampled on a uniform time grid, meaning that irregular or missing timestamps must still be handled in pre‑processing. Furthermore, treating variates as the sequential axis ties runtime and memory directly to the number of channels, which can become a bottleneck for extremely high‑dimensional multivariate time‑series.
> > Moreover, the simultaneous mixing of multiple views blends temporal and cross‑channel information in ways that make detailed attributions difficult and may leave the network sensitive to regime shifts or extremely noisy signals. Addressing these aspects through adaptive variate grouping, continuous‑time embeddings, and lightweight explanation modules are potential paths forward beyond the current work.
>
> > **Societal Impact.**
> > This paper presents work whose goal is to advance the field of Machine Learning. There are many potential societal consequences of our work, none of which we feel must be specifically highlighted here.
> >
> > Our research advances machine learning by enhancing the capabilities of long-term forecasting in time series models, significantly improving both accuracy and efficiency. By developing xLSTM-Mixer, we introduce a robust framework that can be applied across various industries, including finance, healthcare, energy, and logistics. The improved forecasting accuracy enables better decision-making in critical areas, such as optimizing resource allocation, predicting market trends, and managing risk.
> >
> > However, we also recognize the potential risks associated with the misuse of these advanced models. Time series forecasting models could be leveraged for malicious purposes, especially when applied at scale. For example, in the financial sector, adversarial agents might manipulate forecasts to create market instability. In political or social contexts, these models could be exploited to predict and influence public opinion or destabilize economies. Additionally, the application of these models in sensitive domains like healthcare and security may lead to unintended consequences if not carefully regulated and ethically deployed.
> >
> > Therefore, it is essential that the use of xLSTM-Mixer, like all machine learning technologies, is guided by responsible practices and ethical considerations. We encourage stakeholders to adopt rigorous evaluation processes to ensure fairness, transparency, and accountability in its deployment, and to remain vigilant to the broader societal implications of time series forecasting technologies.
>
> Thank you again for your thoughtful review. We hope this review answered all outstanding questions. Specifically, having improved the clarity of the work (theory and writing) and provided impressive new benchmark results on the more practical GIFT-Eval. Furthermore, we added a new limitation section and a more extensive discussion of ablation results.
> Therefore, we have addressed all key concerns regarding our work. In light of this, we would greatly appreciate it if you could raise your score to reflect this.
>
> Best
>
> The Authors
>
> **References:**
> - Zhang et al. A Survey on Multi-Task Learning. IEEE TKDE, 34(12):5586–5609, 2022.
> - Aksu et al. GIFT-Eval: A Benchmark for General Time Series Forecasting Model Evaluation. NeurIPS Workshop on Time Series in the Age of Large Models, 2024.
> - Beck et al. xLSTM: Extended Long Short-Term Memory. NeurIPS, 2024.
> - Greff et al. LSTM: A Search Space Odyssey. IEEE T-NNLS, 2017.
> - Merrill et al. Unlocking State-Tracking in Linear RNNs Through Negative Eigenvalues. NeurIPS Workshop on MML, 2024.
> - Alharthi et al. xLSTMTime: Long-Term Time Series Forecasting with xLSTM. MDPI Sensors, 2024.
> - Auer et al. TiRex: Zero-Shot Forecasting Across Long and Short Horizons with Enhanced In-Context Learning. arXiv:2505.23719, 2025.

---

> > ### Author Response · Authors · 2025-08-04
> > **Any further concerns?**
> >
> > Dear reviewer,
> >
> > Since the discussion period will end in a couple of days, we would like to ask if there are any further questions from your side. We have replied in detail to all your original concerns and hope that we have alleviated your concens. We would be happy to discuss further if necessary.
> >
> > Regards,
> >
> > The authors

---

### Official Review · Reviewer_V19p · 2025-07-02

**Clarity:** 3
**Significance:** 2
**Originality:** 2
**Rating:** 5
**Confidence:** 3

**Summary:**

This paper introduces xLSTM-Mixer, a model for multivariate time series forecasting. It combines a linear forecast with further refinement using xLSTM blocks, effectively integrating temporal sequences and view mixing to capture complex dependencies. The model demonstrates superior long-term forecasting performance compared to previous methods in most cases, while maintaining a very low memory usage. The authors also provide a thorough model analysis, offering insights into the contributions of each component and confirming the model's robustness and effectiveness.

**Questions:**

1. Why sLSTM is more suitable for the mixer framework than the other RNN unit? Is there any theoretical explanation for this phenomenon?

**Ethical Concerns:**

["NO or VERY MINOR ethics concerns only"]

**Final Justification:**

The authors' responses have addressed some of my concerns. More experiment results are provided. I will update my score to 5.

**Limitations:**

The authors only provide limited discussion about the limitation in the Conclusion section.  The discussions about the model's assumptions, data requirements, and failure modes would be beneficial.

**Paper Formatting Concerns:**

None.

**Quality:**

3

**Strengths And Weaknesses:**

Strengths:

1. The paper presents a new model xLSTM-Mixer that combines linear forecasting with xLSTM blocks and a multi-view mixing approach to effectively capture complex patterns in multivariate time series data.

2. xLSTM-Mixer achieves state-of-the-art long-term forecasting results across multiple datasets, outperforming previous methods in the majority of cases.

3. The model maintains a low memory usage and is computationally efficient, making it suitable for deployment on devices with limited resources.


Weakness:

1. The paper focuses primarily on empirical results without providing any theoretical analysis to justify the design choices or explain the model's performance. Theoretical insights into why and how xLSTM-Mixer works could strengthen the contributions.

2. While the paper discusses some limitations of xLSTM-Mixer, a more comprehensive discussion of potential weaknesses would be beneficial. This includes limitations related to the model's assumptions, data requirements, and failure modes.

3. The paper briefly mentions the model's ability for probabilistic forecasting, but the evaluation is limited.

4. The authors acknowledge that the performance of xLSTM-Mixer depends on the ordering of variates, but they only provide limited experiments on this aspect. A more thorough investigation into how different variate orderings affect the model's performance and the development of strategies to optimally determine variate orderings could enhance the model's applicability and effectiveness.

5. The comparisons are limited to existing time series forecasting models. Incorporating more recent state-of-the-art approaches could provide a more comprehensive evaluation.

a) TimeCMA: Towards LLM-Empowered Multivariate Time Series Forecasting via Cross-Modality Alignment. AAAI 2025

b) HDT: Hierarchical Discrete Transformer for Multivariate Time Series Forecasting. AAAI 2025

c) Unlocking the Power of LSTM for Long Term Time Series Forecasting. AAAI 2025

---

> ### Author Rebuttal · Authors · 2025-07-31
>
> Dear Reviewer V19p,
>
> We are grateful for the recognition of the clear writing, the novel xLSTM-Mixer architecture, and excellent modeling capabilities while remaining efficient.
>
> We gladly clarify the raised comments and questions:
>
> **W1:** We ground xLSTM‑Mixer in time‑ and variate‑mixing theory. A single shared NLinear forecaster that marches over the variates (Section 3.1, motivated by Section 2.2) imposes a strong channel‑independent inductive bias. After transposing the embedding—as shown by the inverted‑token iTransformer (Liu et al., 2023)—the sLSTM stack (Section 3.2) jointly mixes time and variates while keeping parameters and memory low. We then produce two complementary latent views (original and reversed), process both with the same sLSTM, and fuse them with one linear layer (Section 3.3); this weight‑tied multi‑view design follows the regularisation principles of multi‑task learning described by Zhang & Yang (2022). Analytically, block‑diagonal recurrence keeps each head focused on long‑range dynamics, while a learnable soft‑prompt token speeds adaptation—a mechanism also used in TEMPO (Cao et al., 2023). Together, these choices explain xLSTM‑Mixer’s state‑of‑the‑art long‑horizon accuracy and efficiency; additional theoretical insights appear in our response to Q1.
>
> **W2:** We welcome the suggestion to improve the manuscript even further by outlining xLSTM-Mixer's limitations. We amended the paper with a new section. Please refer to Reviewer fZrP Response for **Comments on Limitations** for the explicit paragraph.
>
> **W3:** We initially only provided an outlook to probabilistic forecasting.
> We welcome your suggestion, and have since then massively extended this to the novel and large probabilistic GIFT-Eval benchmark (Aksu et al., 2024), as also suggested by Reviewer 67JL.
> In summary, we are pleased to report that xLSTM-Mixer surpasses classical deep-learning and statistical baselines by a large margin on univariate and multivariate datasets, while matching the strongest pre-trained models.
> Please refer to Reviewer 67JL's response for the benchmark tables.
>
> We are grateful for this insightful suggestion, which has broadened the scope of our work and strengthened the manuscript. We will include the full results in the camera-ready version and publish them on the public leaderboard.
>
> **W4:** We agree that this is an intriguing idea for further in-depth analysis.
> These different views can be seen as ensembling with weight-sharing.
> Therefore, extending the size of the ensemble is a natural next step.
> We thus performed additional variate experiments with permutations of the variates beyond one (standard order) and two (standard and reversed). Please refer to Reviewer 67JL's response for the scaling summary tables.
> Again, using two views works best for all prediction lengths, while further increasing the ensemble size can also be beneficial over the single view. However, this may over-regularize the model for some prediction lengths and higher ensemble counts. We observed this on multiple datasets and show ETTh2 as a representative sample. Thank you again for suggesting this insightful perspective. We will include this in our revised manuscript.
>
> **W5:** For the main comparison, we ensured that the comparison was against the *state-of-the-art* in each model group (recurrent, convolutional, etc.). Those are not necessarily the newest models. For instance, PatchTST is still the best model in the "deep-learning" category of the GIFT-eval leaderboard. However, we still include three models from 2025 to reflect the latest developments.
> Regarding the three concurrent works you suggested:
> TimeCMA shows worse performance than xLSTM-Mixer in all 48 comparable settings (Liu et al., 2025, Tab. 1).
> HDT is a model solely for probabilistic forecasting (Shibo et al., 2025), and thus not a natural baseline to compare against.
> Similarly to TimeCMA, xLSTM-Mixer outperforms P-sLSTM in all 24 comparable settings (Kong et al., 2025, Tab. 2).
> However, we gladly incorporate these three recent works into our related work, especially Kong et al. (2025), which is closely related.
> As already discussed in **W3**, we substantially expanded the empirical findings to the GIFT-eval benchmark, thereby adding new baselines via its leaderboard.
>
> **Q1:** Thank you for raising this point.
> Beck et al. (2024) provide an in-depth discussion of why xLSTMs are beneficial over previous recurrent models. Nevertheless, below, we clarify and give an intuition as to why we use sLSTM over mLSTMs. The core reason for that is the explicit state‑tracking via memory mixing in sLSTM, which is lacking from mLSTMs:
>
> - sLSTM keeps the hidden‑to‑hidden (memory mixing) connections of the classic LSTM and augments them with exponential gating and stabilisation (Beck et al. 2024; Greff et al. 2017).
>   $$
>   c_t = f_t c_{t-1} + i_t z_t,\qquad
>   h_t = o_t \, g(c_t)
>   $$
>   Because $f_t$, $i_t$, and $o_t$ all depend on the previous hidden state $h_{t-1}$, the block can learn to blend past and present information on demand.
> - mLSTM stores key–value pairs in a matrix
>   $$
>   C_t = f_t C_{t-1} + i_t v_t k_t^T .
>   $$
>   But to enable full parallelisation, it's gates depend only on the current input, i.e., all hidden‑to‑hidden paths are removed (Beck et al. 2024, Sec 2.3). Thus, memory mixing is absent.
>
> Why this matters for forecasting: Time‑series forecasting often requires the model to keep track of latent phases such as "heating ↔ cooling" or "upward ↔ downward" movement. The model must retain that phase until evidence of a switch appears and flip its internal indicator when the switch occurs.
>
> * In sLSTM, gate values can be trained so that
>   *when a switch indicator is detected* $i_t = 0$, $f_t = 1$ (overwrite),
>   *otherwise* $i_t = 1$, $f_t = 0$ (retain).
>   Because the gates read $h_{t-1}$, the decision depends on the current internal phase as well as the new input, enabling reliable phase tracking.
> * In mLSTM, gate values cannot depend on $h_{t-1}$. Instead, the update is a linear function of the input history, i.e., it can integrate evidence but cannot implement a conditional flip.  Empirically, models without memory mixing (mLSTM, Mamba, Transformers) fail even simple state‑tracking tasks such as parity (Merrill et al. 2024; Beck et al. 2024).
>
> An illustrative Example: Consider a binary series where $x_t = 1$ signals a phase switch; the desired hidden phase $s_t$ obeys $s_t = s_{t-1} \oplus x_t$, where $\oplus$ denotes the XOR operation.
>
> sLSTM case: Choose weights so that $f_t = \sigma(\alpha(1-x_t))$ and $i_t = \sigma(\alpha x_t)$ with large $\alpha$.
> Let the output gate turn $c_t$ into two well‑separated values (e.g., −1 vs 1).
> Then $c_t$ flips sign exactly when $x_t=1$, reproducing $s_t$. Because $f_t$ and $i_t$ both read $h_{t-1}$, more complex phase logic (e.g., multi‑step confirmation) can also be learned.
>
> mLSTM case: Unfolding the recurrence shows that $C_T$ is a weighted sum of outer products of current inputs. In other words, the weights are fixed functions of those inputs alone. Hence, $h_T$ is a linear function of $x_t$. Parity (and any phase‑flip automaton) are not linearly representable—therefore mLSTM cannot track the phase.
>
> Empirical evidence in our work:
> * We employ only sLSTM blocks because mLSTM "treat\[s] sequence elements independently, making it impossible to learn relationships between them directly" (Beck et al. 2024, Sec 2.1).
> * Replacing sLSTM with mLSTM, GRU, or vanilla LSTM consistently degrades long‑horizon accuracy, especially on 336‑ and 720‑step forecasts (cf. Sec 4.4).
> * xLSTMTime uses sLSTM for smaller forecasting datasets where accurate phase tracking dominates, while mLSTM is chosen only when raw storage capacity outweighs this need (Alharthi & Mahmood 2024). TiRex attributes its performance gains to retaining state tracking as well (Auer et al. 2025).
>
> Thus, sLSTM’s memory‑mixing recurrence preserves and updates time‑varying hidden phases, a property indispensable for high‑fidelity forecasts over long horizons. mLSTM's parallel‑friendly design diminishes this ability. Our empirical results and prior literature confirm that, in forecasting contexts, state‑tracking capacity outweighs raw matrix‑memory capacity—hence our exclusive use of sLSTM blocks in xLSTM‑Mixer. We again thank the reviewer for this question and will integrate this much more extensive theoretical view with illustrative examples in the camera-ready version.
>
> Thank you again for your thoughtful review. We hope this review answered all outstanding questions. Specifically, having elaborated on the theory and limitations. Furthermore, we provided impressive leaderboard placements on the more practical and probabilistic GIFT-Eval set, and discussed and validated additional variate ordering findings.
> Hence, we have addressed all key concerns regarding the work. In light of this, we would greatly appreciate it if you could raise your score to reflect this.
>
> Best
>
> The Authors
>
> **References:**
> - Liu et al. TimeCMA: Towards LLM-Empowered Multivariate Time Series Forecasting via Cross-Modality Alignment. AAAI, 39(18):18780–18788, 2025.
> - Shibo et al. HDT: Hierarchical Discrete Transformer for Multivariate Time Series Forecasting. AAAI, 39(1):746–754, 2025.
> - Kong et al. Unlocking the Power of LSTM for Long Term Time Series Forecasting. AAAI, 39(11):11968–11976, 2025.
> - Aksu et al. GIFT-Eval: A Benchmark for General Time Series Forecasting Model Evaluation. NeurIPS Workshop on Time Series in the Age of Large Models, 2024.
> - Zhang et al. A Survey on Multi-Task Learning. IEEE TKDE, 34(12):5586–5609, 2022.
> - Liu et al. iTransformer: Inverted Transformers are Effective for Time Series Forecasting. ICLR, 2024.
> - Cao et al. TEMPO: Prompt-based Generative Pre-trained Transformer for Time Series Forecasting. ICLR, 2023.

---

> > ### Comment · Reviewer_V19p · 2025-08-04
> >
> > Thank you for your response.

---

> ### Author Response · Authors · 2025-08-05
>
> Dear V19p,
>
> We appreciate the time and consideration you’ve given to our submission.
>
> We hope our detailed replies (to you and the other reviewers) have addressed your original concerns thoroughly. If there are any remaining questions or points that you believe warrant further clarification or discussion, we would be glad to engage further.
>
> Please don’t hesitate to let us know if additional input would be helpful.
>
> Regards,
>
> The Authors

---

### Official Review · Reviewer_67JL · 2025-07-03

**Clarity:** 1
**Significance:** 2
**Originality:** 2
**Rating:** 4
**Confidence:** 4

**Summary:**

This paper proposes a new approach for time-serles multivariate forecasting, eg providing point forecast on a group of time-series. The authors propose a novel architecture based on xLSTM which refines a simple linear forecast with sLSTM cells before projecting the output back to the desired dimension. Experiments are conducted on 7 real-world datasets with different forecast horizons to study the effect of long-term forecasting which shows empirical gain against the set of methods compared.

**Questions:**

Here are my questions, sorted roughly by order of importance.

* Regarding baselines selection, the baselines you use are quite outdated, for instance you use Moirai_base and PatchTST which are #14 and #15 in gifteval leaderboard, why not using more recent approaches for instance the latest extension of Chronos or methods performing better on this leaderboard? (which you cite)

* Regarding benchmarking, could you report results on Gifteval leaderboard? This would allow to know better the performance of your method, currently you report results on only 7 datasets (where 4 are ETT variants which are quite related), this may be limited to draw conclusions.

> l605: Hyperparameter tuning was conducted using Optuna [Akiba et al., 2019] with the choices provided in Tab. 4.We optimized for the L1 forecast error, also known as the Mean Absolute Error (MAE).

* Are you tuning the hyperparameters (other than context lengths) only for your method?

* Could you provide a statistical analysis of your results? Are the method statistically tied for instance if you draw a critical diagram? (it seems 7 datasets may not be enough). For instance, it feels that almost all results would be statistically tied between #1 and #6 in your ablation


Those are mostly for curiosity:
- have you considering ensembling different ordering? (right now you just consider two orders, one being reversed)

**Ethical Concerns:**

["NO or VERY MINOR ethics concerns only"]

**Final Justification:**

My main initial concerns were on experiments and clarity:
* Regarding experiments, the author provided a very comprehensive answer that significantly improves the empirical results and analysis.
* Regarding clarity, they mention they will fix the issues I raised but the method description was very poor, while I think it will be improved I have some uncertainty whereas the final description will be clear and with a low number of typos.

For those reasons, I raised my score from 2 to 4. I did not raise to 5 given the uncertainty on the final clarity of the paper which cant be evaluated.

**Limitations:**

Yes

**Quality:**

3

**Strengths And Weaknesses:**

**Quality.** The architecture proposed is well motivated. The authors shows also empirical gains against the set of baselines considered, they also consider many practical scenarios: classification, probabilistic forecasting, different forecast length. However, I have concern about 1) the validity of the claims made due to the lack of statistical analysis 2) the choice of baselines which seems a bit outdated, see questions 3) the hyperparameter optimization protocol which seems to be applied only to their method (despite searching for context lengths).

**Clarity.** The method description is not well written at the moment. While the high-level description is clear, the notations have many issues and inconsistencies which makes it hard to understand all the details.

For instance, the author use bold notations very inconsistently (eg in Eq l98-l99, c_t and h_t are bold and not bold very inconsistently). In 3.1, $\mathbf{x_t}$ is used to denote a single time-series whereas it refers to list of $V$ values the page before. This makes it impossible to me to understand what the authors exactly means for instance in Eq l139 where $x_T^\text{norm}$ has no proper definition. Those imprecise notations are almost everywhere in the model section, for instance l162 $h_0$, l180 $FC^\text{view}$ takes two arguments while having an input space of $\mathbb{R}^D$...

**Significance.** Time-series forecasting is an important applications and as such, the potential impact is significant. One caveat is that the paper use baselines that may be slightly outdated, they also use 7 datasets while more recent benchmarks such as gift-eval typically use a much larger number to ensure significance. One important aspect of the contribution is the gain in inference runtime and memory which is clearly demonstrated.

**Originality.** This paper is one of the first to apply xLSTM to time-series forecasting.

---

> ### Author Rebuttal · Authors · 2025-07-31
>
> Dear Reviewer 67JL,
>
> We thank the reviewer for recognizing the novel xLSTM-Mixer architecture and its empirical gains over the many baselines in multiple practical scenarios.
>
> We gladly clarify the raised comments and questions:
>
> **Re. Clarity:** Thank you for thoroughly checking the notation and verifying its correctness. We already clarified the ambiguous pieces and incorporated the changes you suggested.
> Specifically, we now consistently use the notation introduced in ll. 81ff. This includes the two missing bold notations in Eqs. (1) and (2) and comments regarding Sec. 3.1: There, $\mathbf{x}_t \in \mathbb{R}^V$ intentionally denotes only a single time step of the multivariate time series $\mathbf{X}$, hence the lower-case bold notation introduced before. The definition of $\mathbf{x}^\text{norm}_t$ follows that of the referenced Kim et al. (2022), where $\gamma$ and $\beta$ should also be bold as they are learned separately for each of the $V$ variates.
> In l. 162, we changed the notation to $\mathbf{h}_0$.
> Regarding l. 180, we now correctly define $\operatorname{FC}^\text{view}: \mathbb{R}^D \times \mathbb{R}^D \to \mathbb{R}^H$ in the preceeding line.
> Thank you again for verifying the notation. We are certain this improved the work.
>
> **Q1**: We aim for best comparability with the existing line of research on long-term time series forecasting—such as the long-term benchmarks evaluated by, for example, Chimera from Behrouz et al. (NeurIPS, 2024), PatchTST from Nie et al. (ICLR, 2023), TiDE from Das et al. (TMLR, 2023), and many preceding works.
> Therefore, we primarily focused on the benchmark of the classical seven datasets, each evaluated over four forecast horizons, along with an additional 17 datasets presented in the outlooks in Appendices F, G, and H. For the main comparison, we ensured evaluation against the state-of-the-art within each model category (e.g., recurrent, convolutional), which are not necessarily the most recent models. For instance, PatchTST remains the top-performing model in the "deep learning" category of the GIFT-eval leaderboard. Nevertheless, we also include three models from 2025 to reflect the latest advancements in the field.
>
> **Q2:** Thank you for directing us to GIFT-Eval. The benchmark targets probabilistic zero-shot evaluation, and more than half of its datasets are univariate. Its accompanying pre-training corpus is designed for foundation models rather than classical deep-learning methods, and it focuses on very short horizons, which was initially outside the planned scope of the manuscript. Nevertheless, we ran the benchmark and are pleased to report that xLSTM-Mixer surpasses classical deep-learning and statistical baselines by a large margin on univariate and multivariate datasets, while matching the strongest pre-trained models.
>
> We are grateful for this insightful suggestion, which has broadened the scope of our work and strengthened the manuscript. We will include the full results in the camera-ready version and publish them on the public leaderboard.
>
> ### Overall Leaderboard
> |model|MASE|CRPS|overall_rank|
> |:-------------------|-------:|-------:|---------------:|
> |TiRex|0.724|0.498|1|
> |TEMPO_ensemble|0.862|0.514|2|
> |👉 **xLSTM-Mixer**|0.783|0.515|3|
> |Toto_Open_Base_1.0|0.75|0.517|4|
> |TabPFN-TS|0.771|0.544|5|
> |YingLong_300m|0.798|0.548|6|
> |timesfm_2_0_500m|0.758|0.55|7|
> |YingLong_110m|0.809|0.557|8|
> |sundial_base_128m|0.75|0.559|9|
> |YingLong_50m|0.822|0.567|10|
> |chronos_bolt_base|0.808|0.574|11|
> |chronos_bolt_small|0.822|0.577|12|
> |TTM-R2-Finetuned|0.756|0.583|13|
> |PatchTST|0.849|0.587|14|
> |Moirai_large|0.875|0.599|15|
> |TFT|0.915|0.605|16|
> |YingLong_6m|0.88|0.609|17|
> |Moirai_base|0.901|0.61|18|
> |iTransformer|0.893|0.62|19|
> |Chronos_large|0.87|0.647|20|
> |Moirai_small|0.946|0.65|21|
> |Chronos_base|0.876|0.652|22|
> |Chronos_small|0.892|0.663|23|
> |TimesFM|1.077|0.68|24|
> |VisionTS|0.863|0.755|25|
> |TIDE|1.091|0.772|26|
> |N-BEATS|0.938|0.816|27|
> |DLinear|1.061|0.846|28|
> |DeepAR|1.343|0.853|29|
> |TTM-R2-Zeroshot|1.02|0.873|30|
> |Lag-Llama|1.228|0.88|31|
> |TTM-R1-Zeroshot|1.079|0.891|32|
> |Auto_Arima|1.074|0.912|33|
> |Timer|1.136|0.97|34|
> |seasonal_naive|1|1|35|
> |Auto_Theta|1.09|1.244|36|
> |Naive|1.27|1.591|37|
> |Crossformer|2.574|1.637|38|
> |Auto_ETS|1.212|7.489|39|
>
> ### Multivariate Leaderboard Top 10
> |model|MASE|CRPS|overall_rank|
> |:-------------------|-------:|-------:|---------------:|
> |TEMPO_ensemble|0.752|0.427|1|
> |👉 **xLSTM-Mixer**|0.84|0.477|2|
> |Toto_Open_Base_1.0|0.767|0.484|3|
> |TiRex|0.771|0.49|4|
> |YingLong_300m|0.84|0.519|5|
> |YingLong_110m|0.83|0.52|6|
> |YingLong_50m|0.853|0.53|7|
> |sundial_base_128m|0.778|0.53|8|
> |TabPFN-TS|0.818|0.544|9|
> |timesfm_2_0_500m|0.803|0.553|10|
>
> ### Univariate Leaderboard Top 10
> |model|MASE|CRPS|overall_rank|
> |:-------------------|-------:|-------:|---------------:|
> |TiRex|0.688|0.505|1|
> |TabPFN-TS|0.735|0.544|2|
> |Toto_Open_Base_1.0|0.737|0.545|3|
> |👉 **xLSTM-Mixer**|0.742|0.546|4|
> |timesfm_2_0_500m|0.724|0.549|5|
> |chronos_bolt_base|0.725|0.552|6|
> |chronos_bolt_small|0.739|0.559|7|
> |Moirai_large|0.773|0.572|8|
> |YingLong_300m|0.766|0.573|9|
> |TTM-R2-Finetuned|0.717|0.576|10|
>
> **Q3:** We put in substantial effort to ensure a fair comparison. We proceeded as follows: For most models, the initial publication already comprehensively evaluated the model on these well-known datasets and found appropriate hyperparameters. For xLSTMTime, this was not the case, and we were not able to fully reproduce the results in the paper despite our best efforts (cf. ll. 330f). We still present the better results from Alharthi and Mahmood (2024) so as not to erroneously underestimate the method. For the datasets for which no such results were available (ETTh2, ETTm1, and ETTm2), we ran individual hyperparameter searches like we did for our method to ensure a fair comparison between all models.
>
> **Q4:** We followed the common practice of evaluating the forecast accuracy by counting wins across models and datasets. However, we agree that statistical testing is the more rigorous and desirable procedure to perform this.
> Following the well-known practice of Demšar (2006), we first perform a Friedman test to ensure the model's performances follow different distributions ($p < 10^{-10}$).
> Using Holm's method, we can then perform a Conover post-hoc test while adjusting for the family-wise error rate.
> We must restrict this comparison to one metric and horizon (here, MSE at 96 steps), since the performance results are strongly correlated for one model and dataset pair and would artificially inflate significance if not accounted for.
>
> At a significance threshold of $\alpha = 0.05$, we find that xLSTM-Mixer is statistically significantly better than all other methods, except for xLSTMTime.
> Yet, the difference in average rank is still impressive, with 1.5 for xLSTM-Mixer and 4.0 for xLSTMTime.
> Furthermore, xLSTMTime is statistically inseparable from many other models, namely TimeMixer, TSMixer, ModernTCN, Chimera, PatchTST, and TiDE, in that order.
>
> We will add this setup and the result as an easily digestible Critical Difference (CD) diagram to Sec. 4.1 of the paper.
>
> **Q5 (curiosity):** Thank you so much for the perspective that variate reordering can be seen as ensembling (with weight–sharing). It also sparked our curiosity! We thus performed additional variate experiments with permutations of the variates beyond one (standard order) and two (standard and reversed).
> Again, using two views works best for all prediction lengths, while further increasing the ensemble size can also be beneficial over the single view. However, this may over-regularize the model for some prediction lengths and higher ensemble counts. We observed this on multiple datasets and show ETTh2 as a representative sample. Thank you again for suggesting this insightful perspective. We will include this in our revised manuscript.
>
> ### Scaling Summary: ETTh2 (pred_len=96)
>
> | Ensemble Size | MSE (mean±imp. over #1) | MAE (mean±imp. over #1) |
> |---|---|---|
> | 1 | 0.274+0.00% | 0.330+0.00% |
> | 2 | 0.267-2.55% | 0.329-0.30% |
> | 5 | 0.273-0.36% | 0.333+0.91% |
> | 8 | 0.275+0.36% | 0.336+1.82% |
> | 10 | 0.274+0.00% | 0.336+1.82% |
>
> ### Scaling Summary: ETTh2 (pred_len=192)
>
> | Ensemble Size | MSE (mean±imp. over #1) | MAE (mean±imp. over #1) |
> |---|---|---|
> | 1 | 0.345+0.00% | 0.375+0.00% |
> | 2 | 0.338-2.03% | 0.375+0.00% |
> | 5 | 0.340-1.45% | 0.380+1.33% |
> | 8 | 0.344-0.29% | 0.382+1.87% |
> | 10 | 0.345+0.00% | 0.383+2.13% |
>
> ### Scaling Summary: ETTh2 (pred_len=336)
>
> | Ensemble Size | MSE (mean±imp. over #1) | MAE (mean±imp. over #1) |
> |---|---|---|
> | 1 | 0.377+0.00% | 0.402+0.00% |
> | 2 | 0.367-2.65% | 0.401-0.25% |
> | 5 | 0.374-0.80% | 0.408+1.49% |
> | 8 | 0.373-1.06% | 0.407+1.24% |
> | 10 | 0.372-1.33% | 0.405+0.75% |
>
> ### Scaling Summary: ETTh2 (pred_len=720)
>
> | Ensemble Size | MSE (mean±imp. over #1) | MAE (mean±imp. over #1) |
> |---|---|---|
> | 1 | 0.397+0.00% | 0.426+0.00% |
> | 2 | 0.388-2.27% | 0.424-0.47% |
> | 5 | 0.393-1.01% | 0.430+0.94% |
> | 8 | 0.408+2.77% | 0.439+3.05% |
> | 10 | 0.411+3.53% | 0.441+3.52% |
>
> Again, thank you very much for this constructive feedback. We hope to have sufficiently answered all questions.
> In particular, the main concerns, namely the clarity, the statistical significance of the results, and new impressive results on GIFT-Eval, have substantially improved our manuscript.
> In light of this, we would greatly appreciate it if you reconsidered your score to reflect this.
>
> Best
>
> The Authors
>
> **References:**
> - Behrouz et al. Chimera: Effectively Modeling Multivariate Time Series with 2D State Space Models. NeurIPS, 2024.
> - Nie et al. A Time Series is Worth 64 Words: Long-term Forecasting with Transformers. ICLR, 2023.
> - Das et al. Long-term Forecasting with TiDE: Time-series Dense Encoder. TMLR, 2023.
> - Alharthi et al. xLSTMTime: Long-Term Time Series Forecasting with xLSTM. MDPI AI, 5(3), 2024.
> - Demšar. Statistical Comparisons of Classifiers over Multiple Data Sets. JMLR, 7, 2006.

---

> > ### Comment · Reviewer_67JL · 2025-08-01
> > **Rebuttal answer**
> >
> > I believe all the new experiments and discussion will significantly improve the paper, in particular the results on gift-eval (where the method appear competitive) and the statistical analysis of the results. The justification of the authors on their HPO procedure also makes sense to me. For the ensemble, I believe the result is interesting as it improves the result a bit, it might be worth to include this when running on gifteval.
> >
> > Some of my concern related to clarity remains as it wont be possible to check whether the notation are correct and self-sufficient and there were many typos that in some cases prevented to properly understand the method; I do hope though that the author would be able to carefully fix them so that there is model ambiguity (regarding your point on using the definition of $\mathbf{x}^\text{norm}_t$ follows that of the referenced Kim et al. (2022), I believe it should be introduced as the paper should be self-consistent).
> >
> > Given the significant and convincing experiments done by the authors, I will raise my score accordingly.

---

> ### Author Response · Authors · 2025-08-04
>
> Dear Reviewer 67JL01,
>
> Thank you for the exceptionally thoughtful feedback and for raising your score. Your insights have substantially strengthened our manuscript. We’re integrating your suggestions and have ensured the camera-ready version is typo-free, precise, and fully self-contained.
>
> Bests
>
> The Authors

---

### Comment · Area_Chair_zsru · 2025-08-03
**Discussion period**

Dear Reviewers,

Thank you so much for all your time and effort supporting NeurIPS!

If you haven't yet, please take a moment to read through the author's rebuttal. If the rebuttal addresses your concerns, please acknowledge this and adjust your scores accordingly. If not, please let them know which concerns remain and if you have any follow-up questions. Your thoughtful feedback is crucial for helping the authors improve their work and advancing the field.

I realize this is a busy time and really appreciate your effort.

Best Regards
Area Chair

---

### Author Response · Authors · 2025-08-09
**Thank You for Your Engagement — Post-Rebuttal Summary**

We sincerely thank all reviewers and the AC for their time, constructive feedback, fruitful discussions, and the updated scores. Your insights have helped us improve the clarity, scope, and impact of our work. We will incorporate all suggestions into the camera-ready version.

Below, we briefly summarize the key updates made during the rebuttal phase:

* **Clarified Notation**
* **Added GIFT-Eval Results**. Furthermore, after a longer hyperparameter search, xLSTM-Mixer ranks as a very strong 2nd in both univariate and multivariate leaderboards.
* **Ablation Clarifications**
* **Additional Variate/View Ensembling Experiments**
* **Statistical Significance Analysis**
* **Updated Limitations and Societal Impact section**
* **Theoretical Justification for sLSTM Choice**

We are deeply grateful — your thoughtful input has made our paper even stronger.

Kind regards,

The Authors

---

### Decision · Program_Chairs · 2025-09-17

**Decision:**

Accept (poster)

**Comment:**

This paper proposes a new algorithm for time series multivariate forecasting based on xLSTM which refines a linear forecast across variates with xLSTM blocks before projecting the output back to the desired dimension. The authors have conducted an extensive set of evaluations on different datasets and baseline algorithms, and added the GIFT-Eval during the rebuttal phase. Their results show that xLSTM has promising performance. There were some concerns about quality of writing and clarity of the paper, as well as a discussion of trade-offs and limitations of the approach. The authors have promised to address these issues in the camera-ready. Overall, the reviewers were uniformly positive about the paper.